# Novel LOTUS-domain proteins are organizational hubs that recruit *C. elegans* Vasa to germ granules

Patricia Giselle Cipriani[1,2], Olivia Bay[1], John Zinno[1], Michelle Gutwein[1], Hin Hark Gan[1], Vinay K Mayya[3], George Chung[1], Jia-Xuan Chen[4†], Hala Fahs[2], Yu Guan[1], Thomas F Duchaine[3], Matthias Selbach[4], Fabio Piano[1,2], Kristin C Gunsalus[1,2]*

[1]Center for Genomics and Systems Biology, Department of Biology, New York University, New York, United States; [2]NYU Abu Dhabi Center for Genomics and Systems Biology, New York University Abu Dhabi, Abu Dhabi, United Arab Emirates; [3]Goodman Cancer Research Centre and Department of Biochemistry, McGill University, Montreal, Canada; [4]Max Delbrück Center for Molecular Medicine, Berlin, Germany

*For correspondence: kcg1@nyu.edu

Present address: [†] Institute of Molecular Biology, Mainz, Germany

Competing interests: The authors declare that no competing interests exist.

**Abstract** We describe MIP-1 and MIP-2, novel paralogous *C. elegans* germ granule components that interact with the intrinsically disordered MEG-3 protein. These proteins promote P granule condensation, form granules independently of MEG-3 in the postembryonic germ line, and balance each other in regulating P granule growth and localization. MIP-1 and MIP-2 each contain two LOTUS domains and intrinsically disordered regions and form homo- and heterodimers. They bind and anchor the Vasa homolog GLH-1 within P granules and are jointly required for coalescence of MEG-3, GLH-1, and PGL proteins. Animals lacking MIP-1 and MIP-2 show temperature-sensitive embryonic lethality, sterility, and mortal germ lines. Germline phenotypes include defects in stem cell self-renewal, meiotic progression, and gamete differentiation. We propose that these proteins serve as scaffolds and organizing centers for ribonucleoprotein networks within P granules that help recruit and balance essential RNA processing machinery to regulate key developmental transitions in the germ line.

## Introduction

The germ line is essential to the propagation of sexually reproducing multicellular organisms. A common feature of germ cells is the presence of germ granules, membraneless cytoplasmic compartments with liquid-like properties that form by liquid-liquid phase separation (LLPS) from the bulk cytoplasm (*Brangwynne et al., 2009*). Rich in both RNA and RNA-binding proteins (RBPs), germ granules contain a large number of molecules with roles in post-transcriptional RNA regulation and the preservation of genome integrity (*Mayya and Duchaine, 2019*). These complex ribonucleoprotein (RNP) assemblies are important for the specification, maintenance, and normal development of the germ line (*Knutson et al., 2017*).

In *C. elegans* adults, germ cells develop along an assembly line within a syncytium, migrating away from the stem cell niche as they proliferate through several rounds of mitosis, enter meiosis, and differentiate into gametes; mature oocytes are then fertilized as they pass through a sperm storage organ called the spermatheca. Polarization of the embryo upon fertilization triggers differential anterior-posterior phosphorylation of multiple regulatory proteins, resulting in concentration gradients that provide a permissive environment for germ granule condensation only in the posterior (*Brangwynne et al., 2009*; *Seydoux, 2018*). *C. elegans* germ granules are called P granules because

during embryogenesis they segregate uniquely to the posterior cell lineage (P0-P4 and Z2/Z3), those cells that will give rise to the future germ line (*Updike and Strome, 2010*). P granules are cytoplasmic in very early embryos, begin to coalesce around the nuclear periphery at the four-cell stage, and thereafter form perinuclear foci in all germ cells (*Updike and Strome, 2010*). Although many P granule components are known, their regulation, organizational principles, and mechanisms through which they influence germline development are still not completely understood.

Nucleation of P granules in the zygote requires the intrinsically disordered, serine-rich protein MEG-3 (previously known as GEI-12) (*Chen et al., 2016*; *Wang et al., 2014*). A putative null allele of *meg-3* is not sterile on its own, but does show sterility when combined with mutations in its paralog *meg-4* (30%), *meg-1* (40%), or both (100%) (*Wang et al., 2014*). Transgenerational RNAi of *meg-3* gives rise to progressive loss of fertility (*Chen et al., 2016*), presumably due to additional off-target knock-down of *meg-4*, which encodes a protein that is 71% identical to MEG-3 (*Wang et al., 2014*). MEG-3 binds RNA non-specifically and recruits numerous mRNAs into P granules (*Lee et al., 2020*), while a MEX-5 gradient suppresses granule formation in the anterior by competing with MEG-3 for binding to RNAs (*Smith et al., 2016*). In vitro, MEG-3 also forms gel-like condensates that are enhanced by the presence of RNA and antagonized by MEX-5 (*Lee et al., 2020*; *Putnam et al., 2019*). Thus, MEG-3 is part of a regulatory network that controls P granule formation in early embryos.

Phase separation of proteins with intrinsically disordered regions (IDRs) is a hallmark of germ granules and other non-membranous organelles that carry out diverse core cellular processes, including gene transcription, ribosome biogenesis, cytoskeletal organization, stress responses, and synaptic activity, and many others (*Alberti and Hyman, 2021*; *Strom and Brangwynne, 2019*; *Wiegand and Hyman, 2020*; *Zhao and Zhang, 2020*). Many IDRs can drive phase separation in vitro and in vivo and make nonspecific, low-affinity, multivalent contacts with RNAs and other proteins that synergize with high-affinity interaction domains to form extended interaction networks (*Banani et al., 2017*; *Peran and Mittag, 2020*; *Protter et al., 2018*; *Sanders et al., 2020*). These 'biomolecular condensates' are complex coacervates that are governed by principles of polymer physics and can undergo phase transitions between liquid-like, gel-like, and more solid, semi-crystalline states (*Brangwynne et al., 2015*; *Shin and Brangwynne, 2017*). Whereas condensed liquid droplets undergo rapid local motions, hydrogels contain fibrils of stacked beta-sheets that can harden into irreversible amyloids that are linked to the etiology of various diseases, most notably neurological and age-related pathologies (*Alberti and Hyman, 2021*). In vivo, 'dissolvases' such as protein kinases and ATP-driven RNA helicases are engaged to maintain fluidity, and these active processes prevent solidification as condensates age (*Hubstenberger et al., 2013*; *Nott et al., 2015*; *Rai et al., 2018*).

To gain further insight into the mechanisms underlying P granule biology, we sought to identify novel P granule proteins that co-immunoprecipitate with known regulators of P granule assembly. Here we describe the characterization of two novel paralogs discovered by co-immunoprecipitation with MEG-3, which we have named '**M**EG-3 **i**nteracting **p**roteins' MIP-1 and MIP-2. Both MIPs contain OST-HTH or LOTUS domains (named for their presence in **L**imkain B1/MARF1, **O**skar, and **Tu**dor domain-containing protein**s** TDRD5 and TDRD7) and are the first proteins of this class to be characterized in *C. elegans*. Oskar is the key initiator of germ plasm assembly in *Drosophila* and recruits the Vasa DEAD-box RNA helicase to germ granules (*Lehmann, 2016*). Here we show that the MIPs likewise recruit a *C. elegans* Vasa homolog, can form multivalent protein-protein interactions, and are essential for proper granule assembly and many aspects of germline development. We propose that the MIPs form extended interaction networks that scaffold and help nucleate, organize, and regulate macromolecular assemblies essential to core germ granule processes.

## Results

### Novel LOTUS-domain proteins interact with MEG-3 in vivo

In a previous study, we found that MEG-3 and MEG-4 co-immunoprecipitate from *C. elegans* early embryo extracts with MBK-2, a master regulator of the oocyte-to-embryo transition (*Chen et al., 2016*). To identify additional proteins that interact with MEG-3 in vivo, we performed co-immunoprecipitation (co-IP) of a GFP::MEG-3 fusion protein from mixed-stage early embryos followed by

tandem mass spectrometry (MS) (*Figure 1A*, *Supplementary file 1a*), using an optimized protocol for label-free quantitative interaction proteomics from that study. To evaluate assay quality and reproducibility, we confirmed that both MEG-4 and MBK-2 were pulled down by MEG-3, demonstrating that these three proteins are found together when using either MEG-3 or MBK-2 as bait. In addition, numerous other known P granule components were present among the MEG-3 interactors (*Figure 1A*).

Two previously uncharacterized paralogous proteins, C38D4.4 and F58G11.3, were strongly enriched and highly significant (*Figure 1A*). Like MEG-3, these proteins contain extensive serine-rich, disordered regions that comprise 60–65% of their length (*Figure 1B*). A single major isoform for each protein was detected in vivo using 3xFLAG-tagged proteins expressed in mixed-stage animals

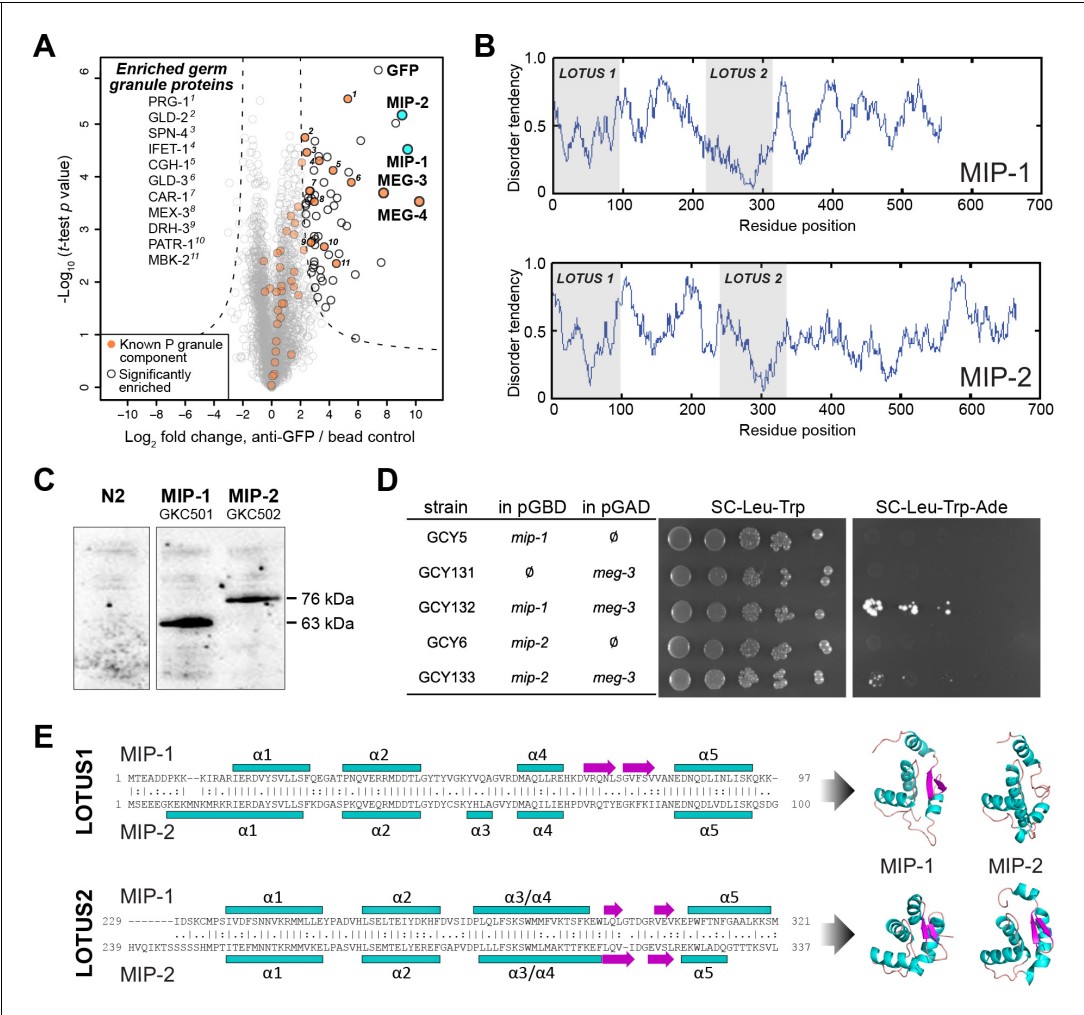

**Figure 1.** MIP-1 and MIP-2 are novel LOTUS-domain proteins. (**A**) Proteins that co-immunoprecipitate with MEG-3 from embryo extracts include known components of P granules (orange), as well as two previously uncharacterized paralogs (aqua): C38D4.4 (MIP-1) and F58G11.3 (MIP-2). Proteins significantly enriched over controls in three biological replicates are outlined in black and include numerous known P granule components (listed at left, in order of descending significance). (**B**) MIP-1 and MIP-2 contain two LOTUS domains and intrinsically disordered regions (IDRs). (**C**) MIP-1 and MIP-2 are detected as a single isoform in extracts of mixed-stage animals. Western hybridization using an anti-FLAG antibody detects one polypeptide for 3xFLAG fusion proteins in genome-edited strains GKC501 (MIP-1::3xFLAG) and GKC502 (MIP-2::3xFLAG). N2 control is from the same gel. (**D**) Both MIP-1 and MIP-2 interact directly with MEG-3 in yeast two-hybrid assays. (**E**) (Left) Sequence alignments of LOTUS domains in MIP-1 and MIP-2. LOTUS domains are predicted to contain an alpha-5 helix that has been found in extended LOTUS domains (eLOTUS). (Right) Predicted three-dimensional structures of LOTUS1 (top) and LOTUS2 (bottom) domains in MIP-1 and MIP-2.

The online version of this article includes the following figure supplement(s) for figure 1:

**Figure supplement 1.** Sequence alignments and 3D structural models for MIP LOTUS domains.

(*Figure 1C*). Both proteins interact directly with MEG-3 in yeast two-hybrid assays (*Figure 1D*). We have named the corresponding genes *mip-1* (C38D4.4) and *mip-2* (F58G11.3).

A combination of primary sequence analysis and predictions of secondary and tertiary structure revealed the presence of two conserved ~100-residue globular LOTUS domains in the N-terminal half of each MIP, separated by a mostly disordered region of ~120–140 residues (*Figure 1E*, *Figure 1—figure supplement 1*). LOTUS domains are found in all kingdoms of life and occur in proteins with diverse domain architectures (*Anantharaman et al., 2010*; *Callebaut and Mornon, 2010*). Bioinformatic analyses and experimental evidence indicate that LOTUS domains mediate specific protein-protein interactions and/or bind RNA. Two LOTUS variants have been distinguished: the minimal LOTUS (mLOTUS) and the extended LOTUS (eLOTUS), whose defining feature is an additional C-terminal alpha-helix in a region that appears disordered in minimal LOTUS domains (*Jeske et al., 2017*). A recent study has identified binding of RNA G-quadruplex secondary structures as an ancient conserved function of mLOTUS domains across kingdoms, and suggests that protein binding by eLOTUS domains may be a more recent evolutionary innovation (*Ding et al., 2020*).

Superpositions of LOTUS domain models from worm, fly, and human display clear structural similarity (*Figure 1—figure supplement 1*, *Supplementary file 1b,c*). Although multiple alignments of LOTUS domains from diverse species show very low sequence identity (10–16%) (*Figure 1—figure supplement 1A*), the ten most conserved residues are predicted to form part of the hydrophobic core and thus would be expected to help stabilize the tertiary structure (*Figure 1—figure supplement 1B*). Homology models of all four MIP LOTUS domains contain a clearly identifiable α5 helix (*Figure 1—figure supplement 1C*) and can thus be classified as eLOTUS domains like that in the *Drosophila* germline protein Oskar.

## MIP-1 and MIP-2 influence multiple aspects of germline development

All metazoan LOTUS proteins studied so far are critical for normal germ cell development. In *Drosophila*, Oskar and the Tudor domain proteins Tejas and Tapas (homologs of mammalian TDRD5 and TDRD7) recruit Vasa to germ granules in developing egg chambers, where they directly bind and stimulate its activity to regulate translation of developmentally important mRNAs or promote piRNA amplification (*Jeske et al., 2015*; *Jeske et al., 2017*). In the nuage in nurse cells, Vasa is essential for the Ping-Pong piRNA amplification system, where it helps drive the exchange of piRNA intermediates between Argonaute proteins (*Xiol et al., 2014*). *Drosophila* MARF1 contains six mLOTUS domains and regulates oocyte maturation by recruiting the CCR4-NOT complex, which regulates polyadenylation, to inhibit translation of a cyclin mRNA (*Zhu et al., 2018*). Mouse MARF1, which contains eight mLOTUS domains, displays mLOTUS-dependent RNA cleavage activity and protects against retrotransposons during oocyte development (*Yao et al., 2018*).

To investigate the in vivo functions of MIP-1 and MIP-2, we performed RNA interference (RNAi) experiments targeting each of the paralogs individually and together. Single-gene RNAi depletions elicited no evident phenotypes, whereas simultaneous depletion of both *mip* genes produced sterile animals at 25˚C. Continuous RNAi by feeding at 25˚C over three generations resulted in progressively smaller brood sizes and embryonic lethality in the progeny of the non-sterile worms, with the F2 generation showing almost complete sterility (*Figure 2A*).

We next deleted the full coding sequence (CDS) of both genes and generated 4x back-crossed homozygous lines carrying the single and double null mutant alleles (referred to below as *mip-1Δ* and *mip-2Δ*). While all three strains appeared superficially wild type (WT), the double null displayed a strong mortal germ line (Mrt) phenotype, resulting in complete sterility after five generations at 20˚C (*Figure 2—figure supplement 2C*). Upon shifting to 25˚C, all three strains displayed a rapid increase in sterility over just two generations: between F1 and F2 generations, sterility in the individual *mip-1Δ* and *mip-2Δ* strains increased from ~3% to 10–15%, and in the *mip-1Δ;mip-2Δ* from ~50% to 70–80%.

We observed a diverse range of specific germline phenotypes that progressively increased in frequency and severity across several generations in both *mip-1*(RNAi);*mip-2*(RNAi) treated animals (*Figure 2B,D*) and *mip-1Δ;mip-2Δ* lines at 25˚C (*Figure 2—figure supplement 2A*). Phenotypes ranged from minor defects to almost complete lack of detectable germline tissue and included defects in mitotic proliferation, progression from mitosis to meiosis, pachytene exit, the switch from spermatogenesis to oogenesis, and the formation of normal gametes (*Supplementary file 1d*). A small proportion of animals displayed an additional 'Gogo' phenotype (germ line-oocyte-germ line-

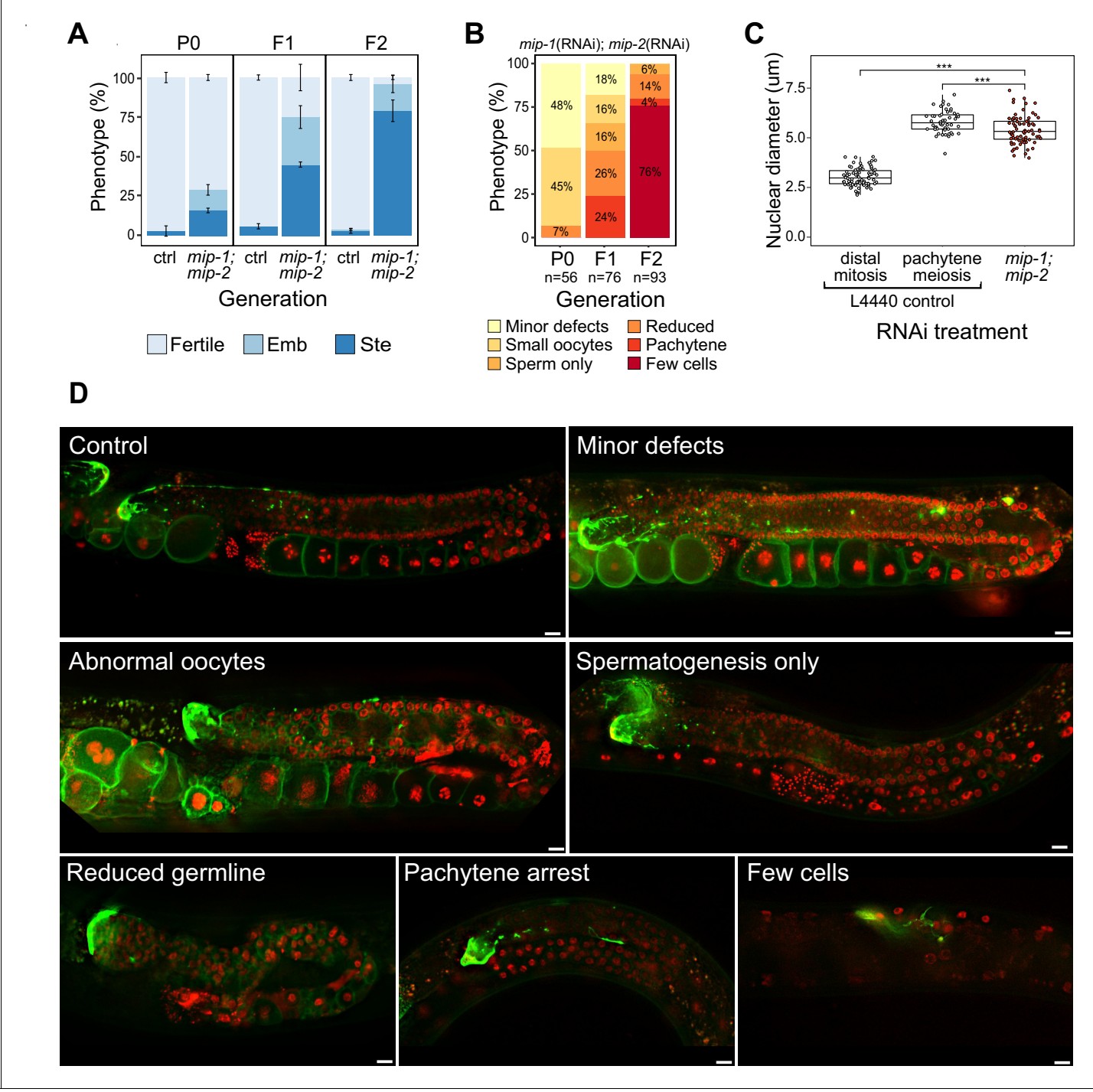

**Figure 2.** Phenotypes caused by reduction of MIP-1 and MIP-2 in the adult germ line. Quantification and examples of increasing phenotypic severity across P0, F1, and F2 generations upon continuous *mip-1(RNAi);mip-2(RNAi)* versus L4440(RNAi) controls. (**A**) Proportion of treated animals at each generation showing embryonic lethality (Emb) and sterility (Ste). Error bars denote SEM for a total of n=~150 independent observations per treatment. (**B**) Proportion of germline phenotypic classes over three generations. A and B show combined data from strains OD95 and GKC509. (**C**) Diameter of remaining germline nuclei (μm) in severely reduced germ lines ('few cells' class) compared with nuclei from controls in strain GKC509. All WT nuclei measured in the distal germ line were within two cell diameters of the DTC. Asterisks denote significance according to Student's *t*-test (***, p<0.001). (**D**) Fluorescent micrographs showing examples of main phenotypic classes in strain GKC509. Scale bars: 10 μm. Strain OD95 carries markers for histone H2B (mCherry::HIS-58) and plasma membrane (GFP::PH(PLCΔ1)); strain GKC509 also carries a marker for the distal tip cell (DTC) (LAG-2::GFP). The online version of this article includes the following source data and figure supplement(s) for figure 2:

**Source data 1.** Phenotypes produced by simultaneous depletion of MIP-1 and MIP-2 by RNAi.

*Figure 2 continued on next page*

*Figure 2 continued*

**Figure supplement 1.** Schematic of protocol for RNAi treatment and scoring of phenotypes for data in *Figure 2* (see also Materials and methods).
**Figure supplement 2.** Double *mip* null mutants show a similar array of germline phenotypes as double *mip* RNAi.
**Figure supplement 2—source data 1.** Double *mip* null mutants show a mortal germline phenotype.

oocyte) (*Eberhard et al., 2013*; *Sendoel et al., 2019*), resembling a duplication of the oogenic program (*Figure 2—figure supplement 2A,B*). Distal oocytes occur in the absence of excess proximal proliferation, and we observed no proximal tumors in any of the germ lines we examined. This spectrum of phenotypes suggests that developmental switches regulating cell state transitions become compromised in germ lines where MIPs function is strongly reduced or abrogated.

The most extreme germline phenotype consisted of only a few germ cells in close proximity to the DTC (*Figure 2D*; *Figure 2—figure supplement 2A*). Germ cell size and nuclear DNA morphology are normally highly stereotypic within each region of the germ line (*Hirsh et al., 1976*). The diameters of the remaining germ cell nuclei measured around 1.8 times larger than those in the distal proliferative zone of control germ lines (*Figure 2C*). While more similar in size to pachytene nuclei in controls, they showed very different DNA morphology. The unusual appearance and small number of remaining nuclei suggest that they have lost their capacity for self-renewal, and we infer that such germ lines have exhausted their pool of stem cells.

## MIP-1 and MIP-2 localize to P granules

To investigate their subcellular localization, we made CRISPR lines carrying fluorescently tagged versions of MIP-1 and MIP-2. MIP-1 is omnipresent throughout development, localizing exclusively to the germline precursor lineage (from P0 through P4) and remaining prominent in the germline progenitor cells Z2 and Z3 (*Figure 3A*, *Figure 3—figure supplement 1A*). In contrast, MIP-2 begins to dissipate in P3 (*Figure 3A*, *Figure 3—figure supplement 1A*) and is virtually undetectable when Z2 and Z3 are born (*Figure 3A*). As P granules coalesce on the nuclear periphery, MIP-1 often appears to localize in closer proximity to the nuclear membrane than MIP-2 (*Figure 3A*, P3 cell; *Figure 3—figure supplement 1B*, P2 cell) and also forms small perinuclear puncta on its own (*Figure 3A*, *Figure 3—figure supplement 1A,B*). In the adult hermaphrodite germ line, MIP-1 is perinuclear through meiosis and also forms cytoplasmic granules in oocytes; MIP-2 shows a similar pattern, but it becomes more diffuse in the cytoplasm around the mitosis/meiosis switch until late pachytene (*Figure 3B*). During spermatogenesis in L4 hermaphrodites and in males, both MIPs form puncta in spermatocytes, are eliminated into residual bodies, and are undetectable in mature sperm (*Figure 3C*). We did not detect MIP-1 or MIP-2 in any somatic tissues.

To investigate if the MIP granules correspond to P granules, we generated homozygous strains carrying fluorescently tagged versions of the MIPs together with selected P granule proteins. Images of fixed animals and time-lapse recordings showed that the MIPs colocalize in granular structures with the constitutive P granule component PGL-1 in both embryos (*Figure 4A*) and the adult germ line (*Figure 4B*) and with MEG-3 in the embryonic P lineage (*Figure 4C,D*). Super-resolution microscopy revealed that both MIP-1 and MIP-2 permeate the full volume of P granules in early embryos (*Figure 4A,B,D*; *Figure 4—figure supplement 1*), as does MEG-3 (*Figure 4D*, *Figure 4—figure supplement 1*). As previously shown for MEG-3 (*Wang et al., 2014*), some MIP-1 granules in the embryo do not include or contain a much lower level of PGL-1 (*Figure 4A*). These are generally localized toward the anterior in P cells, where PGL proteins are thought to be disassembled more rapidly than other P granule components like MEG-3 (*Wang et al., 2014*). Thus, MIP-1 and MIP-2 accumulate in P granules and show largely overlapping localization patterns throughout development.

## MIP-1 and MIP-2 are required for normal P granule assembly

P granule formation is severely compromised when *mip-1* and *mip-2* are simultaneously depleted by RNAi, as revealed by confocal imaging of PGL-3, GLH-1 and MEG-3 granules (*Figure 5A–C*, *Figure 5—videos 1–3*). Time-lapse recordings of RNAi-treated embryos revealed that PGL-3, GLH-1, and MEG-3 appeared mostly diffuse in the zygote; remaining puncta were smaller and less intense, and condensation of all three components was delayed (*Figure 5—videos 1–3*). Residual PGL

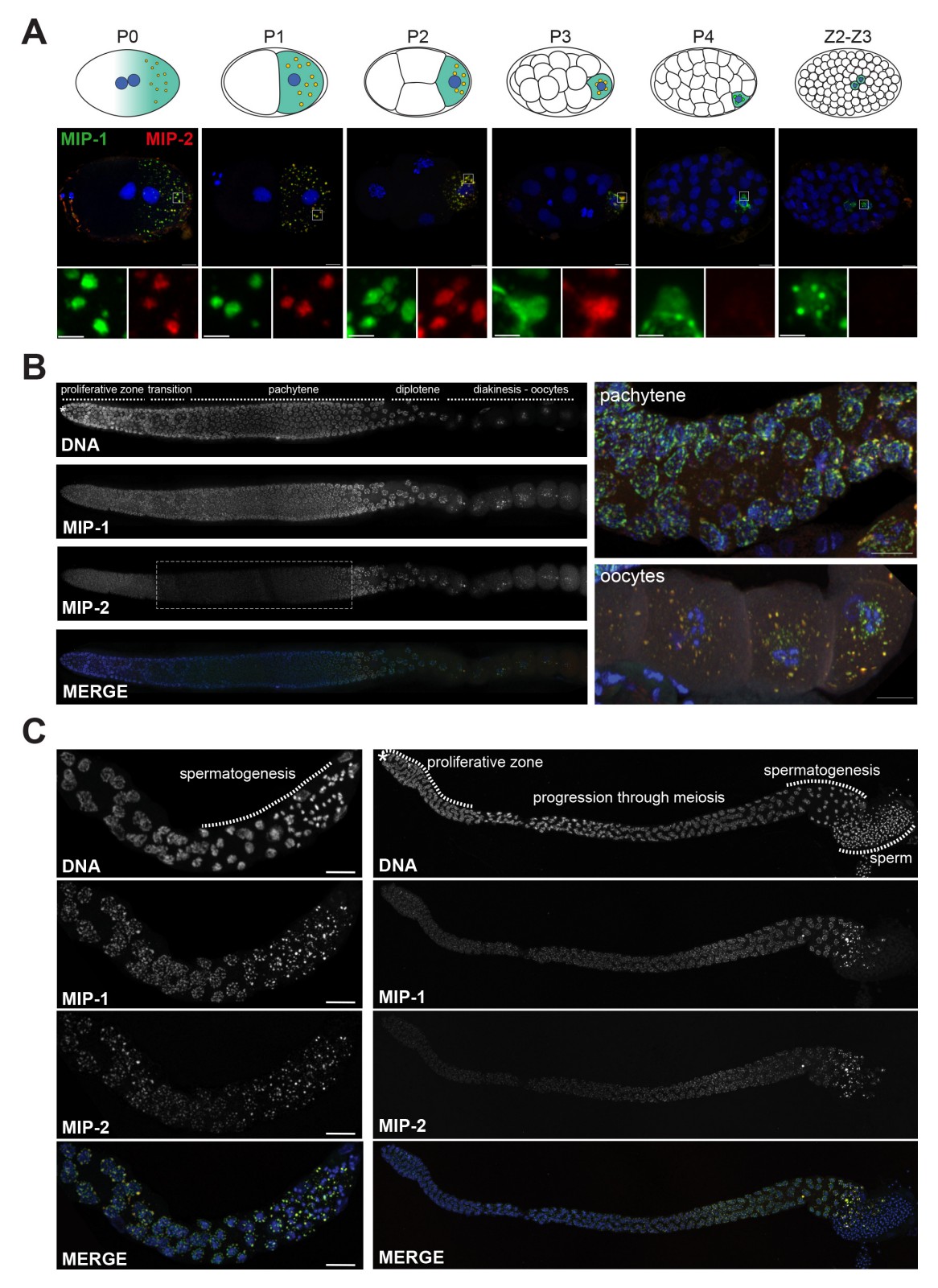

**Figure 3.** MIP-1 and MIP-2 form granules in the germ cell lineage and germ line throughout development. Micrographs of fixed samples harboring MIP-1::GFP and MIP-2::mCherry. (**A**) MIP-1 and MIP-2 colocalize in granular structures in the germline precursor lineage in the embryo; beginning in P4 the expression of MIP-2 begins to decrease, becoming undetectable in older embryos (Z2–Z3). Cartoons of embryo developmental stages highlighting P lineage blastomeres (top panels), corresponding super-resolution micrographs (middle panels), and insets showing magnified sections from the

*Figure 3 continued on next page*

*Figure 3 continued*

embryos above in separate channels (bottom panels). Green puncta, GFP-tagged MIP-1, red puncta mCherry-tagged MIP-2, yellow color denotes the presence of similar levels of both proteins. Scale bar: 2 µm. (B) MIP-1 and MIP-2 localize to granules around germ cell nuclei in the germ line. Epifluorescence images of dissected gonads (left panel) and Zeiss LSM880 Airyscan images of pachytene nuclei and oocytes (right panel). MIP-1 is evenly expressed throughout the adult germ line, while MIP-2 expression is attenuated between the proliferative zone and pachytene exit (dashed box). (C) MIP-1 and MIP-2 are expressed during spermatogenesis in late L4 hermaphrodites (left panel) and adult males (right panel). Airyscan images of dissected gonads are oriented with the distal germ line to the left. Scale bar for left-hand panels:10 µm.

The online version of this article includes the following figure supplement(s) for figure 3:

**Figure supplement 1.** MIP-1 and MIP-2 colocalize in live embryos.

granules also failed to concentrate in the posterior of P lineage cells prior to mitosis and were mis-segregated to the sister cell (*Figure 5—video 1*). Very similar anterior disassembly defects in the early embryo also occur upon depletion of MEG-3 and MEG-4 (*Ouyang et al., 2019*; *Wang et al., 2014*). It is likely that *mip-1Δ;mip-2Δ* null embryos carrying these markers would show even stronger defects; however, we were unable to maintain these lines due to their high level of sterility. In the adult germ line, PGL-3 and GLH-1 granules were largely dispersed upon depletion of *mip-1* and *mip-2*, with only a few small puncta remaining (*Figure 5B,C*).

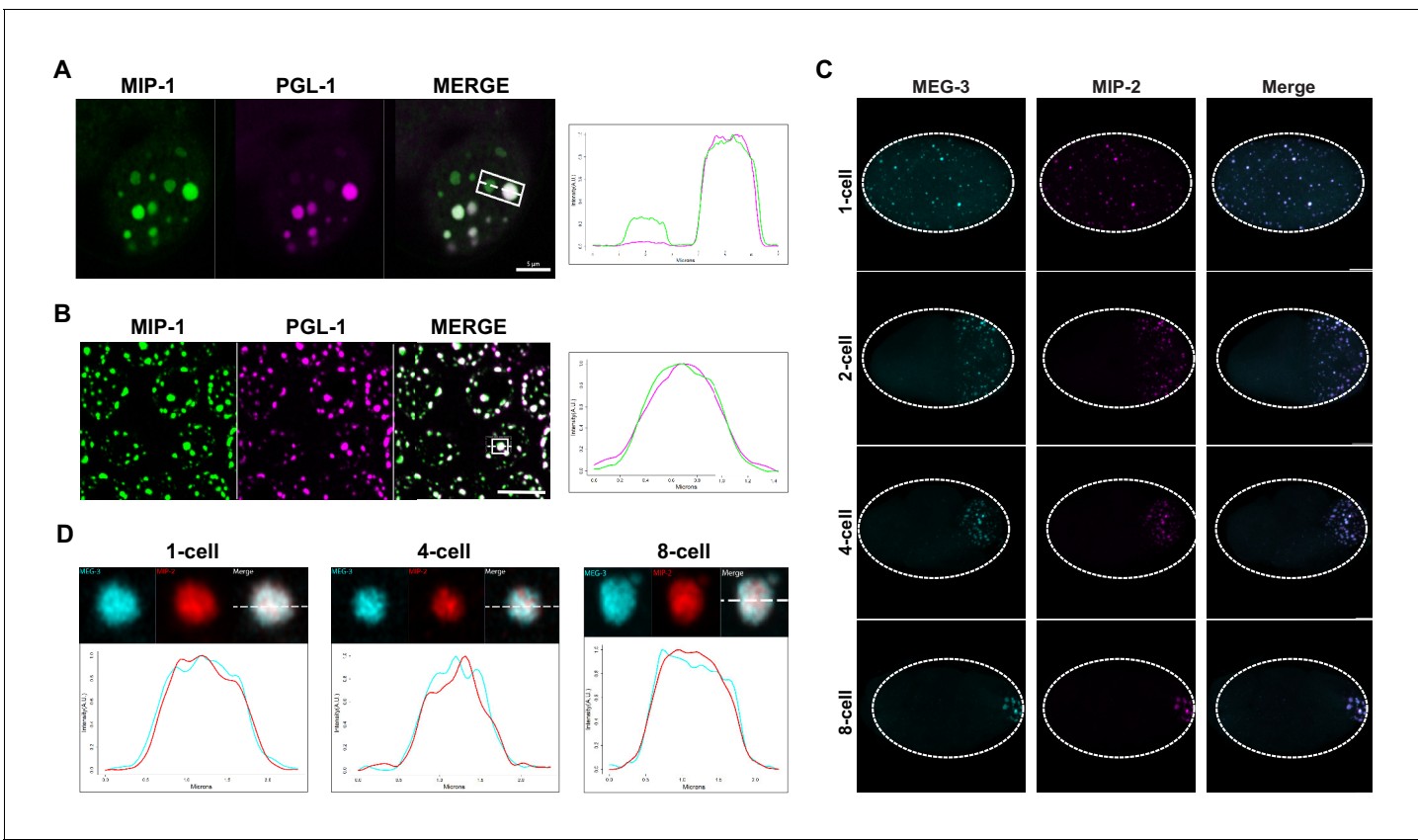

**Figure 4.** MIP-1 and MIP-2 are P granule components. (A-B) MIP-1::GFP (green) and PGL-1::mCherry (fuschia) colocalize in (A) embryos and (B) adult germ line. (A) Left: P granules in P2 cell of a live four-cell embryo (anterior to the left). Some anteriorly localized granules contain MIP-1 and not PGL-1. Right: Cross-sectional fluorescence profile of the two granules highlighted in the micrograph, showing MIP-1::GFP and PGL-1::mCherry expression. (B) Left: Perinuclear P granules in the pachytene region. Right: Fluorescence profile of the granule highlighted in the micrograph. (C) Colocalization of MEG-3::Cerulean and MIP-2::mCherry in the early embryo. (D) Fluorescence profiles of granules from 1-cell, 4-cell, and 8-cell embryos showing complete overlap of MEG-3::Cerulean (aqua) and MIP-2::mCherry (red). Scale bars, 5 µm.

The online version of this article includes the following figure supplement(s) for figure 4:

**Figure supplement 1.** MIP-2 and MEG-3 colocalize throughout the volume of P granules in the early embryo.

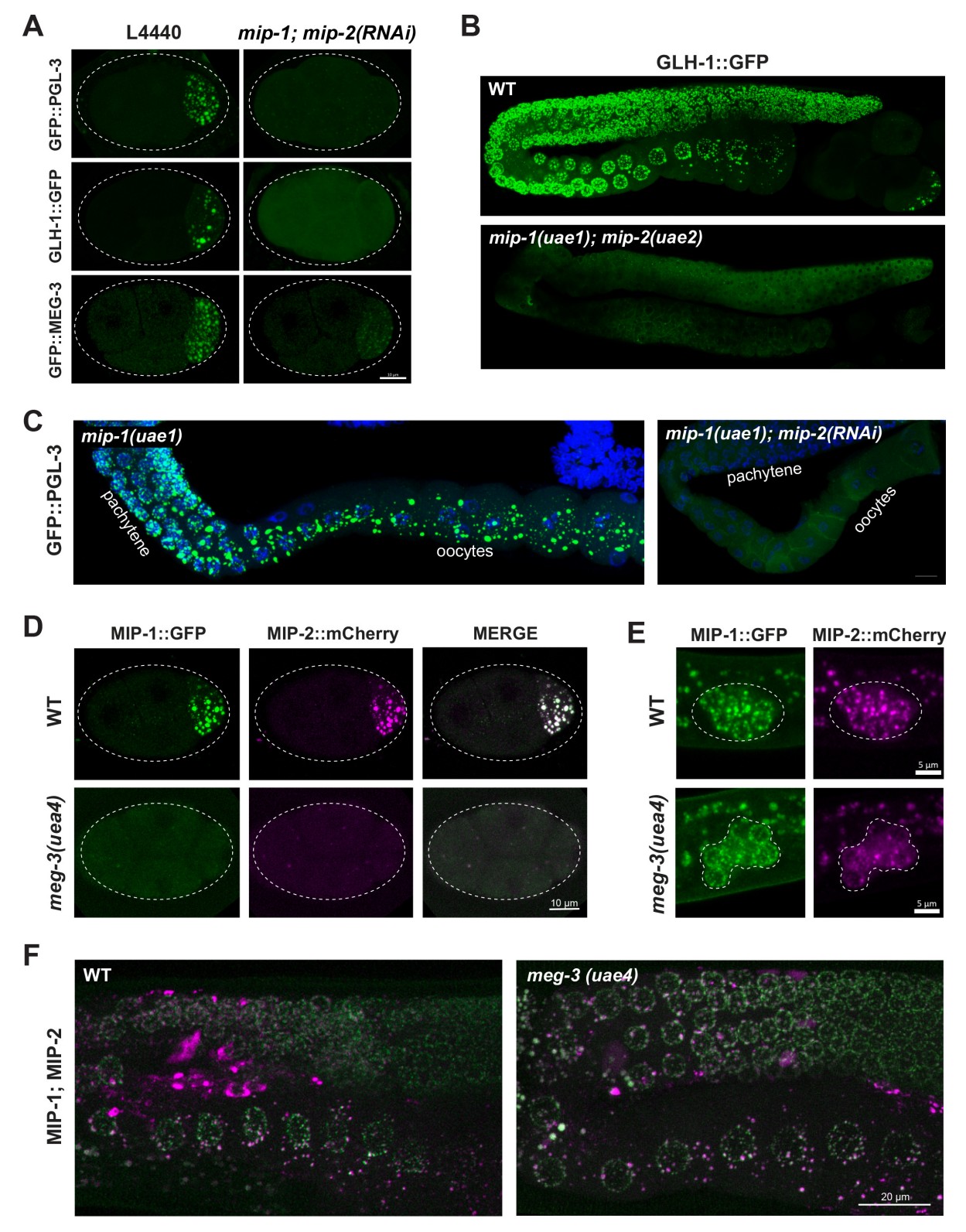

**Figure 5.** MIP-1 and MIP-2 are required for assembly of core P granule components. (**A**) Simultaneous depletion of MIP-1 and MIP-2 by RNAi affects the normal coalescence of GFP::PGL-3, GLH-1::GFP, and GFP::MEG-3 granules in the embryonic P lineage. Embryos shown are at the four-cell stage (see also *Figure 5—videos 1–3*). (**B**) GLH-1::GFP is mostly diffuse, with few small visible puncta, in the germ line of a live *mip-1(uae1);mip-2(uae2)* double null animal (strain GKC555). (**C**) Dissected and fixed germ lines carrying GFP::PGL-3 in a *mip-1(uae1)* genetic background (strain GKC525) show

*Figure 5 continued on next page*

*Figure 5 continued*

nearly complete loss of PGL-3 granules in the F1 generation when treated with *mip-2(RNAi)* (right) but not control RNAi (empty vector, left). (D–E) MIP-1::GFP and MIP-2::mCherry localization in WT and *meg-3(uae4)* null embryos (D) and L1 larvae (E). Deletion of *meg-3* impairs MIP granule formation in embryos (four-cell embryos shown here) but not in L1 larvae, which express MIP-1 and MIP-2 at the onset of germ cell proliferation in dividing germ cells (enclosed by dashed line). Bright puncta outside the germ line in the green channel are autofluorescent intestinal gut granules. In both A and D, embryos in mutant backgrounds are over-exposed to visualize the blastomeres. (F) MIP-1 and MIP-2 expression in the adult germ line (WT, left) is unaffected in the presence of a *meg-3* null allele (right). Dissected germ lines were stained with DAPI to visualize DNA.

The online version of this article includes the following video and figure supplement(s) for figure 5:

**Figure supplement 1.** Localization of MIP-1 and MIP-2 in the adult germ line is not dependent on *pgl-1*, *pgl-3*, or *meg-3*.

**Figure 5—video 1.** Normal formation of PGL-3 granules is affected in the early embryo when *mips* are depleted.

https://elifesciences.org/articles/60833#fig5video1

**Figure 5—video 2.** Normal formation of GLH-1 granules is affected in the early embryo when *mips* are depleted.

https://elifesciences.org/articles/60833#fig5video2

**Figure 5—video 3.** Normal formation of MEG-3 granules is affected in the early embryo when *mips* are depleted.

https://elifesciences.org/articles/60833#fig5video3

**Figure 5—video 4.** Normal formation of MIPs granules is affected in the early embryo of a *meg-3* null mutant.

https://elifesciences.org/articles/60833#fig5video4

**Figure 5—video 5.** MEG-3 localizes to P granules in the embryo but not in the adult germ line.

https://elifesciences.org/articles/60833#fig5video5

MIP-1 and MIP-2 appeared mostly diffuse and formed only a few small granules in early embryos homozygous for a *meg-3* null allele (*Figure 5D*, *Figure 5—video 4*) but MIP granules remained visible in early embryos depleted of *pgl-1* or *pgl-3* (*Figure 5—figure supplement 1A*). We observed no obvious differences from WT in localization of either MIP in larval or adult germ lines of animals that hatched from *meg-3* null embryos (*Figure 5E,F*), consistent with the absence of detectable MEG-3 protein in the postembryonic germ line of WT animals (*Figure 5—figure supplement 1B* and *Figure 5—video 5*). MIP-1 and MIP-2 localization in the adult germ line also appeared largely unaffected by RNAi depletion of *pgl-1, pgl-3* and *meg-3* (*Figure 5—figure supplement 1A*). Thus, the MIPs and MEG-3 are mutually co-dependent for proper P granule assembly in embryos, and other core P granule components are strongly dependent on MIP-1 and MIP-2 for proper localization at all developmental stages.

## Biophysical properties of MIP-1 and MIP-2

Liquid-liquid phase separation of some biomolecular condensates depends primarily on hydrophobic interactions, and these readily dissolve upon exposure to the aliphatic alcohol 1,6-hexanediol (*Updike et al., 2011*). Previous studies have shown that fluorescently tagged PGL-1, PGL-3, and GLH-1 rapidly lose their granular appearance in the presence of hexanediol, whereas MEG-3 granules partially resist this treatment (*Putnam et al., 2019*; *Updike et al., 2011*). Fluorescence recovery after photobleaching (FRAP) experiments have shown that PGL-3 also recovers from photobleaching more rapidly than MEG-3 in very early embryos (*Putnam et al., 2019*). Based on these analyses, PGL and MEG-3 proteins have respectively been described as liquid and gel phases of embryonic P granules (*Putnam et al., 2019*).

We tested the behavior of the MIPs using both methods. Quantification of GFP-tagged MIP and GLH-1 granules in dissected adult germ lines showed that MIP (but not GLH-1) granules are partially resistant to 1,6-hexanediol treatment (*Figure 6A*), and thus their retention within perinuclear granules is not fully dependent on hydrophobic interactions. FRAP assays of GFP-tagged proteins showed that PGL-3 recovered more rapidly than GLH-1, MIP-1, MIP-2, and MEG-3 in the P1 cell of two-cell embryos, when granules are very mobile and free floating in the cytoplasm (*Figure 6B*, left). In contrast, all five proteins recovered at similar rates in the P3 cell in 8- to 12-cell embryos (*Figure 6B*, right), when granules are mostly attached to the nuclear membrane. Thus, PGL proteins are more dynamic than other core P granule components in the cytoplasmic P granule condensates of very early embryos but these differences largely disappear over time, indicating that the physical properties of these condensates evolve as they begin to coalesce around the nuclear periphery. Notably, all proteins tested appear to be fully exchangeable with a cytoplasmic pool in early

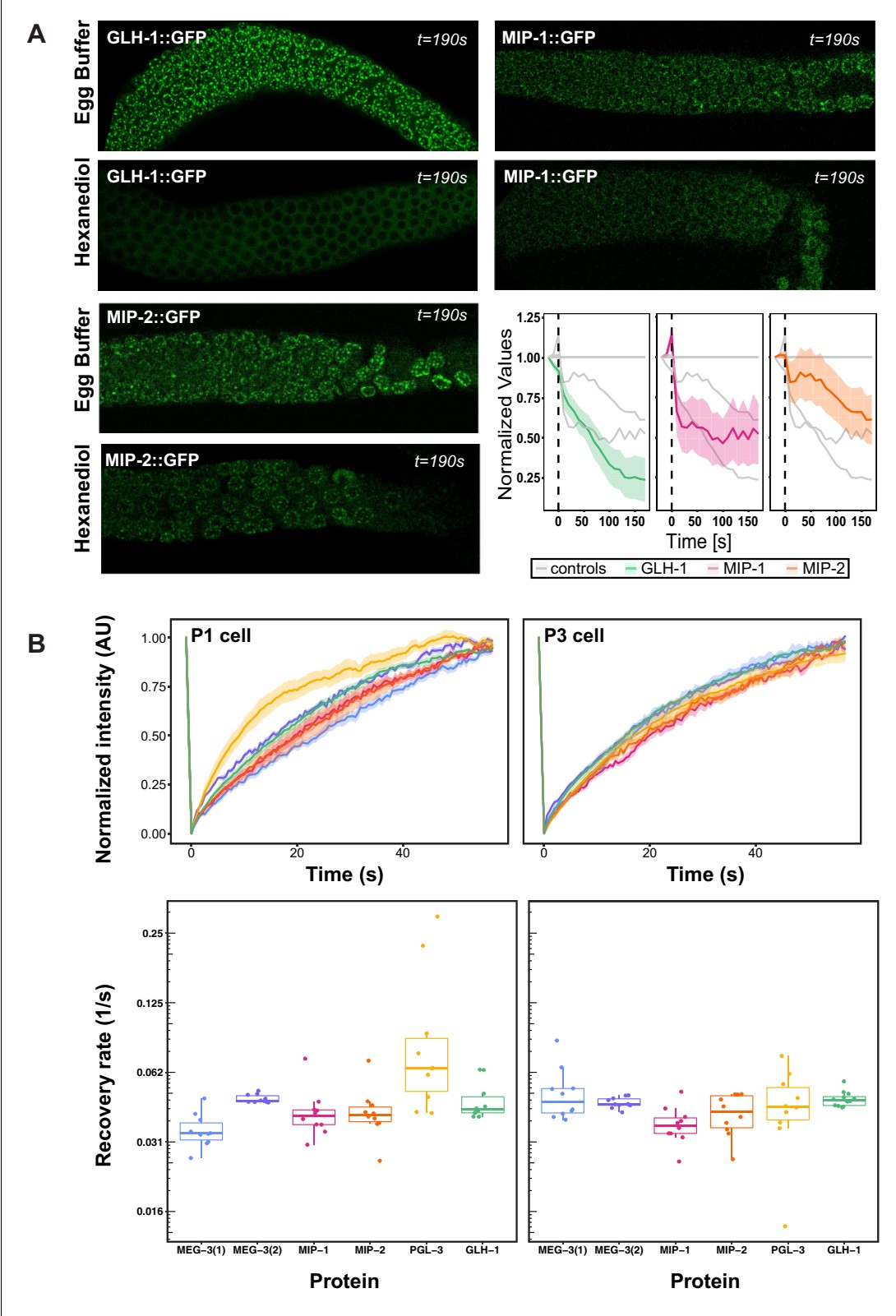

**Figure 6.** Biophysical properties of P granule components. (**A**) Dissected MIP-1::GFP, MIP-2::GFP, and GLH-1::GFP adult germ lines treated with 5% 1,6-hexanediol (HD) or egg buffer as a control (EB), imaged at the last experimental time point (190 s post-treatment). Quantification of the relative number of perinuclear granules over time (bottom right panel) indicates that MIP-1 and MIP-2 granules are less sensitive than GLH-1 to disruption of hydrophobic interactions. (**B**) Top: FRAP of individual P granules in P1 cells of two-cell (left) and P3 cells of 8- to 12-cell (right) embryos carrying

*Figure 6 continued on next page*

*Figure 6 continued*

individual GFP-tagged P granule proteins. Curves show mean normalized fluorescence intensity and standard error over time (n=10 replicates each). Bottom: Quantification of recovery rates from data shown in top row (median and interquartile range). PGL-3 recovery rate is higher than the other proteins in two-cell embryos; statistical differences are negligible between all other measurements (ANOVA test). Recovery rates were measured in two different MEG-3 GFP-tagged strains made by (1) bombardment [strain JH3016] or (2) CRISPR [strain JH3503].

The online version of this article includes the following source data for figure 6:

**Source data 1.** Biophysical properties of MIPs and other germ granule proteins in the germ line and early embryos.

embryos, as normalized intensity in FRAP experiments returns to baseline levels within around 60 s (*Figure 6B*; *Ishikawa-Ankerhold et al., 2012*).

## MIP-1 and MIP-2 physically interact

Dimerization of Oskar through its LOTUS domain is essential for the formation of functional germ plasm in *Drosophila* (*Jeske et al., 2015*). To assess whether the MIPs might interact similarly, we first used the solved structure of the Oskar LOTUS dimer as a template to assemble and refine 3D models of all possible MIP-1 and MIP-2 LOTUS homo- and heterodimers. Predicted binding affinities for several of these combinations were favorable, indicating the potential feasibility of these complexes (*Supplementary file 1e*). We next performed in vitro pull-down assays using purified bacterially expressed full-length proteins, N- and C-terminal regions, and individual LOTUS domains (*Figure 7*). We found that both MIP-1 and MIP-2 N-terminal regions (each containing two LOTUS domains) can interact with full-length MIP-1 (*Figure 7C,D*), and that all four individual LOTUS domains were able to pull down full-length MIP-1 (*Figure 7E,F*). Therefore, MIP-1 can both self-associate and bind MIP-2 through interactions involving their LOTUS domains. The dual-LOTUS MIP-1 N-terminal region bound full-length MIP-1 more strongly than either individual LOTUS domain in vitro, suggesting that LOTUS domain interactions may be additive or cooperative (*Figure 7E*). We found that the IDR-containing C-terminal region of MIP-2 also binds full-length MIP-1 (*Figure 7D*). Further inspection of this region revealed a small predicted helical bundle spanning residues ~462–575. We do not yet know whether this region contributes nonspecific and/or specific interactions that increase overall binding affinity between the two proteins.

## MIPs directly bind and recruit *C. elegans* Vasa to P granules

Oskar both dimerizes and binds the C-terminal domain (CTD) of Vasa through its eLOTUS domain. Oskar binding stimulates Vasa's helicase activity through an interaction surface opposite to the eLOTUS dimer interface (*Jeske et al., 2015*; *Jeske et al., 2017*; *Yang et al., 2015*). The tertiary structure of this complex revealed that the eLOTUS α5 helix directly interacts with Vasa and is crucial for stabilizing this interaction. We superimposed the Vasa (CTD) 3D structure with a model of the GLH-1 (CTD) and found that they are nearly identical (1.4 Å backbone RMSD over 416 residues). We then used the Vasa (CTD)-Oskar (eLOTUS) 3D structure as a template to assemble and refine hypothetical GLH-1(CTD)-LOTUS complexes for all four MIP-1 and MIP-2 LOTUS domains. The resulting 3D models suggested that the LOTUS1 domains in both MIP proteins can potentially bind GLH-1 (*Supplementary file 1e*, *Figure 8*).

We investigated these predictions experimentally using both in vitro co-IP and yeast two-hybrid (Y2H) assays. Both assays showed binding between full-length MIP-1 and GLH-1 (*Figure 8B–D*). Pull-down assays further showed that full-length MIP-1 can bind the region of GLH-1 containing its two helicase domains (aa 301–763). Y2H assays, though not quantitative, showed a weaker interaction between MIP-2 and full-length GLH-1. Assays with swapped GAL4 DNA-binding and activation domains were negative, possibly due to steric hindrance.

We reasoned that if GLH-1 binds to MIP-1 in vivo, depleting MIP function should have an effect on the mobility of GLH-1. Indeed, GLH-1::GFP granules in P3 cells of *mip-1* null embryos showed a much faster recovery rate in FRAP experiments (*Figure 8E*). We conclude from these data that both MIP-1 and MIP-2 bind the GLH-1 CTD domain directly through their LOTUS domains and thus could potentially compete with each other for binding in vivo.

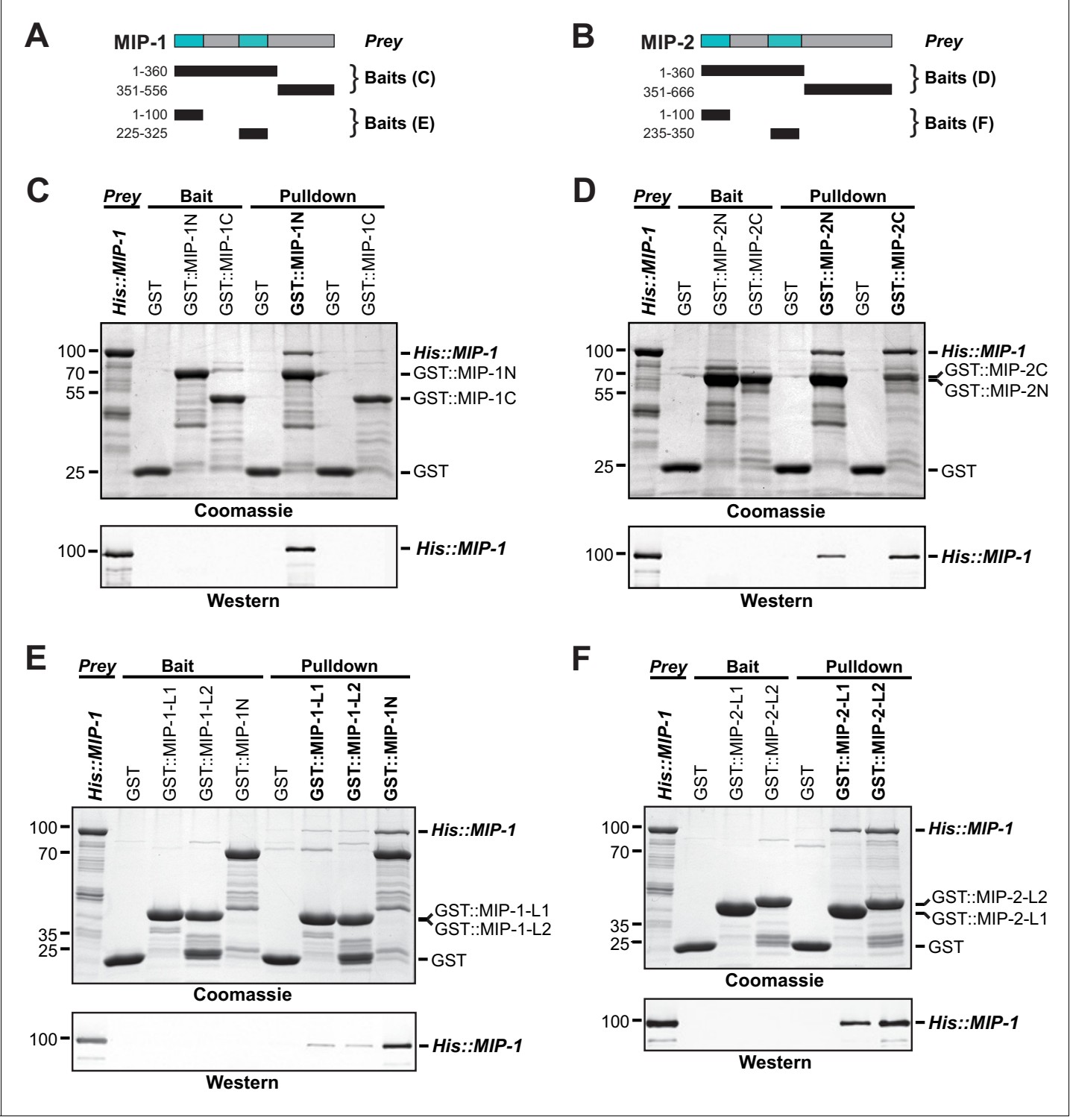

**Figure 7.** MIP-1 and MIP-2 physically interact. (A-B) Cartoons indicating prey and bait protein fragments used in co-immunoprecipitation experiments. LOTUS domains are depicted in cyan. (C–E) Co-immunoprecipitation of full-length 6xHis-tagged MIP-1 with GST-tagged N- and C-terminal fragments of MIP-1 (C) or MIP-2 (D) LOTUS1 and LOTUS 2 of MIP-1 (E) or LOTUS1 and LOTUS 2 of MIP-2 (F) purified recombinant proteins. (C) MIP-1 homodimerizes through its N-terminal region. (D) Full-length MIP-1 interacts with both the N- and C-terminal fragments of MIP-2. (E–F) Individual MIP-1 (E) and MIP-2 (F) LOTUS1 and LOTUS2 domains interact with full-length MIP-1.

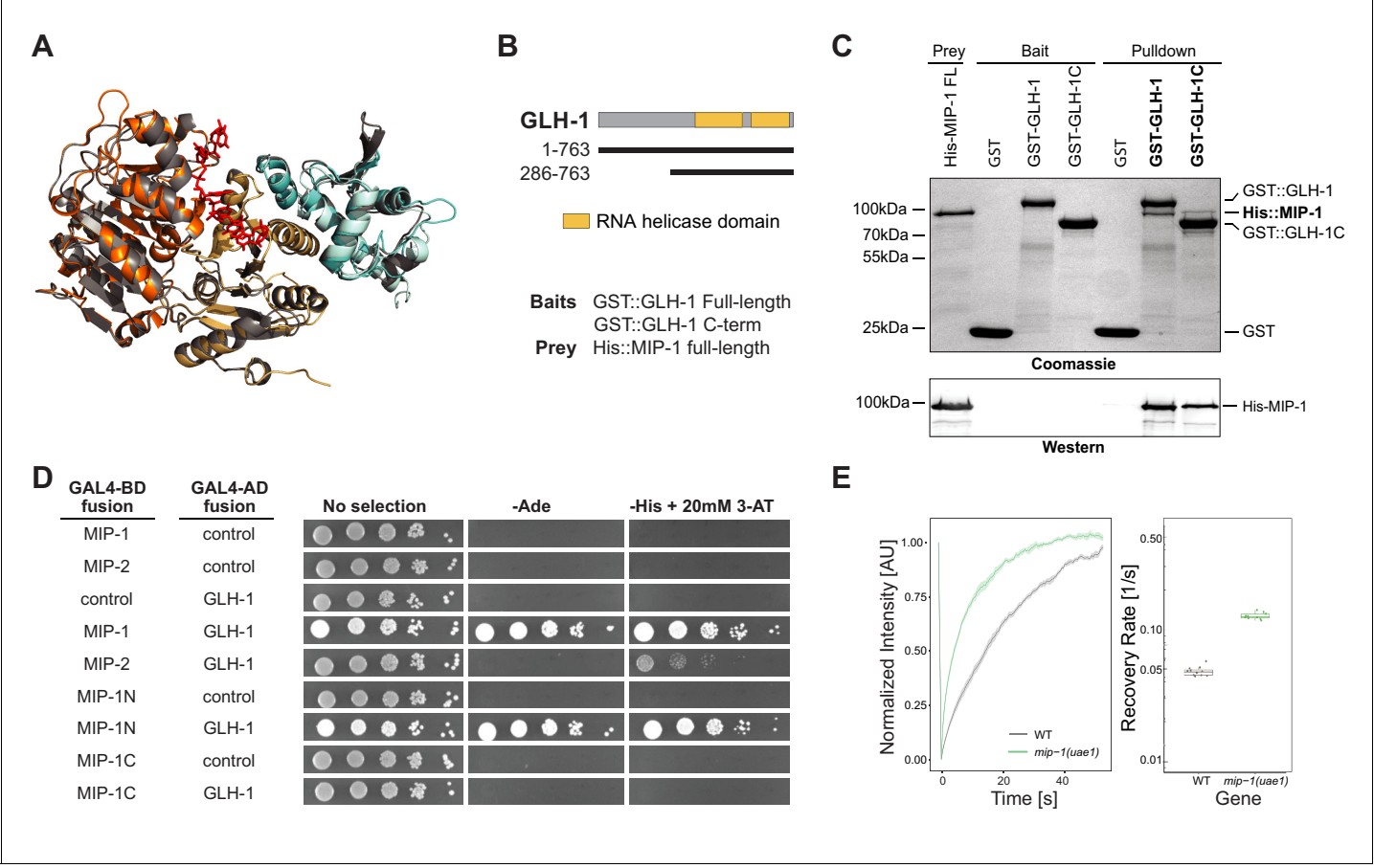

**Figure 8.** MIP-1 and MIP-2 interact directly with GLH-1, a Vasa helicase ortholog. (**A**) 3D structural models showing the overlap between Vasa helicase bound to the Oskar LOTUS domain (gray ribbons) and GLH-1 helicase N-terminal (orange) and C-terminal (salmon) helicase domains bound to LOTUS1 of MIP-1 (cyan) and MIP-2 (sky blue). (**B**) Schematic of GLH-1 constructs used. The N- and C-terminal helicase domains (orange) span residues 372–556 and 592–739 and are separated by a short linker. (**C**) His-tagged full-length MIP-1 co-immunoprecipitates with GST-tagged full-length GLH-1 and with the GLH-1 C-terminal region. (**D**) Yeast two-hybrid assays for MIP-1 and MIP-2 interaction with GLH-1. Yeast Gal4 DNA-binding domain (Gal4-BD) and Gal4 activation domain (Gal4-AD) were fused to *C. elegans* MIP constructs containing a terminal 3xFLAG tag. Positive interactions display prototrophy upon activation of ADE2 and HIS3 reporters. Full-length MIP-1 and residues 1–360, spanning its two LOTUS domains, interact directly with GLH-1. MIP-2 also interacts with GLH-1 but gives a weaker result by this assay. (**E**) FRAP curves and recovery rates of GLH-1::GFP in P3 cells of (gray) WT and (green) *mip-1(uae1)* null embryos (n=10 replicates each). The recovery rate of GLH-1::GFP is significantly increased in the absence of MIP-1, indicating increased mobility.

The online version of this article includes the following source data for figure 8:

**Source data 1.** The mobility of GLH-1 granules depends on the presence of MIP-1 in P3 cells.

## MIP-1 and MIP-2 balance each other in granule formation

To investigate whether MIP-1 and MIP-2 influence each other's localization in vivo, we generated strains carrying a fluorescently tagged version of one gene in a genetic background harboring a homozygous deletion allele of the other (***Figure 9***). In the adult germ line, MIP-2 largely lost its association with the nuclear periphery in a *mip-1* null mutant background and formed larger granules in the rachis and in the cytoplasm of oocytes. Conversely, in a *mip-2* null mutant background, MIP-1 granules in oocytes were reduced in number and size, and remaining granules appeared in closer proximity to the nucleus. In early embryos, the effects on granule size appeared similar, but were less pronounced (***Figure 9—figure supplement 1***, ***Figure 9—videos 1 and 2***).

We also examined the localization of other P granule components in the adult germ line in individual *mip-1* or *mip-2* null backgrounds. GLH-1::GFP (***Figure 10***, ***Figure 10—video 1***) and GFP::PGL-3 (***Figure 10—figure supplement 1***) showed similar changes in localization patterns in combination with individual *mip* deletion alleles: formation of large cytoplasmic granules in the absence of

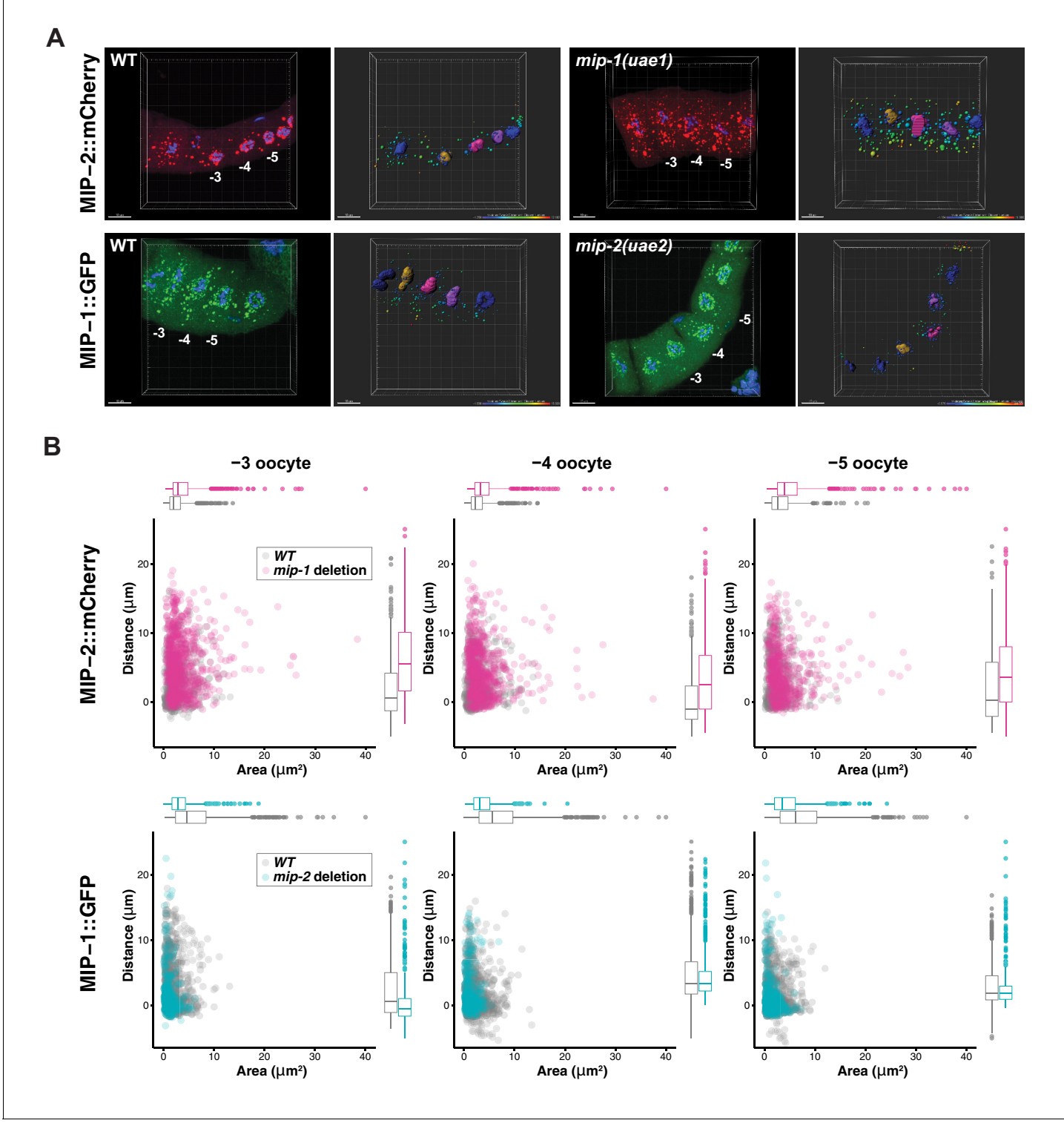

**Figure 9.** MIP-1 and MIP-2 have opposing effects on granule growth and distribution in the adult germ line. (**A**) Identification of granules in the −3, −4, and −5 oocytes using Imaris image analysis software. Examples of typical micrographs of dissected germ lines used as input for the software are paired with their corresponding Imaris outputs for WT (left) and homozygous *mip* deletion alleles (right). Oocyte nuclei are colored according to position from the uterus: −3 (amber), −4 (fuchsia), −5 (purple). Other nuclei not included in the analysis are colored in blue. Identified granules are depicted using a color gradient according to their distance from nuclear boundaries (blue, closest; red, farthest away). Top row: MIP-2::mCherry in WT vs. *mip-1(uae1)* null background, showing increased size P granules in the cytoplasm. Granules are also seen in the rachis. Bottom row: MIP-1::GFP in WT vs. *mip-2*

*Figure 9 continued on next page*

*Figure 9 continued*

(*uae2*) null background, showing concentration of P granules around the nuclear periphery and reduction of cytoplasmic granule number and size. Scale bars, 10 µm. (B) Quantification of granule area and distance from nuclei in −3, −4, and −5 oocytes (10 gonads per genotype).

The online version of this article includes the following video, source data, and figure supplement(s) for figure 9:

**Source data 1.** MIP-1 and MIP-2 affect each other's localization and granule size in the germline.
**Figure supplement 1.** MIPs affect each other's condensation in embryos.
**Figure supplement 1—source data 1.** MIP-1 and MIP-2 affect each other's granule size in early embryos.
**Figure 9—video 1.** MIP-2 granule formation depends on *mip-1* in the early embryo.
https://elifesciences.org/articles/60833#fig9video1
**Figure 9—video 2.** MIP-1 granule formation depends on *mip-2* in the early embryo.
https://elifesciences.org/articles/60833#fig9video2

MIP-1 and fewer cytoplasmic granules in the absence of MIP-2. We analyzed GLH-1::GFP localization in both fixed dissected gonads (*Figure 10A*) and in live animals (*Figure 10B*, *Figure 10—video 1*). In *mip-1Δ* homozygotes, we observed that the GLH-1::GFP signal was also reduced in the pachytene region. Since MIP-2 itself is reduced in this region in WT animals, and localization of GLH-1 to P granules is MIP-dependent, these results are consistent with expectation.

We conclude that MIP-1 is important for tethering P granules to the nuclear periphery, consistent with its appearance closer to the base of perinuclear P granules in WT germ lines, and that the MIPs balance each other to regulate the overall size and distribution of P granule condensates within germ cells (*Figure 11*).

## Discussion

In this work, we describe the first phenotypic and molecular characterization of *C. elegans* LOTUS domain proteins. The novel paralogs MIP-1 and MIP-2 are required to maintain fertility across generations, and their depletion gives rise to pleiotropic germline phenotypes, including defects in germline stem cell maintenance, meiotic progression, and oocyte development. We show that the MIPs are core P granule components that promote germ granule condensation at all stages of development and function at an early step in P granule assembly. The MIPs directly bind and are co-dependent with the embryo-specific protein MEG-3 for granule condensation in the early embryo, and they balance each other's function in regulating P granule size and distribution. The MIPs can form multivalent interactions and also directly bind and anchor the Vasa homolog GLH-1 within P granules, a property they share with other germline LOTUS proteins including Oskar, the key initiator of germ granule assembly in *Drosophila*.

### MIP molecular structure and function

Our combined data strongly suggest that the MIPs form higher order complexes in vivo through a combination of homotypic and heterotypic dimerization, LOTUS-helicase interactions, and low-specificity IDR contacts with other proteins and/or RNAs. MIP-1 and MIP-2 can self-associate and form heterodimers through their LOTUS domains, and MIP-2 also interacts with MIP-1 through its largely disordered C-terminal region. Although we do not know if MIPs bind RNAs directly, this is a distinct possibility given the tendency of many IDRs to bind RNA and the evolutionary conservation of G-quadruplex binding among LOTUS domains (*Ding et al., 2020*).

Several observations lead us to conclude that MIP-1 and MIP-2 are not completely redundant paralogs. First, phenotypic analysis shows that the MIPs have opposing effects on P granule size and localization in vivo. To our knowledge, this is the first example of pair of paralogs that balance each other's function in this way, in germ granules or other types of biomolecular condensates. In addition, Y2H results, although not quantitative, suggest that MIP-1 could bind GLH-1 more strongly than MIP-2, and only MIP-1 LOTUS1 is predicted to bind GLH-1 with high affinity based on structural modeling. While it remains to be determined if MIP-1 and MIP-2 bind other germline helicases and whether they stimulate helicase activity in a manner similar to Oskar and Vasa, our data suggest that they could regulate P granule dynamics in part by engaging helicases to provide reaction centers for a range of RNP regulatory complexes.

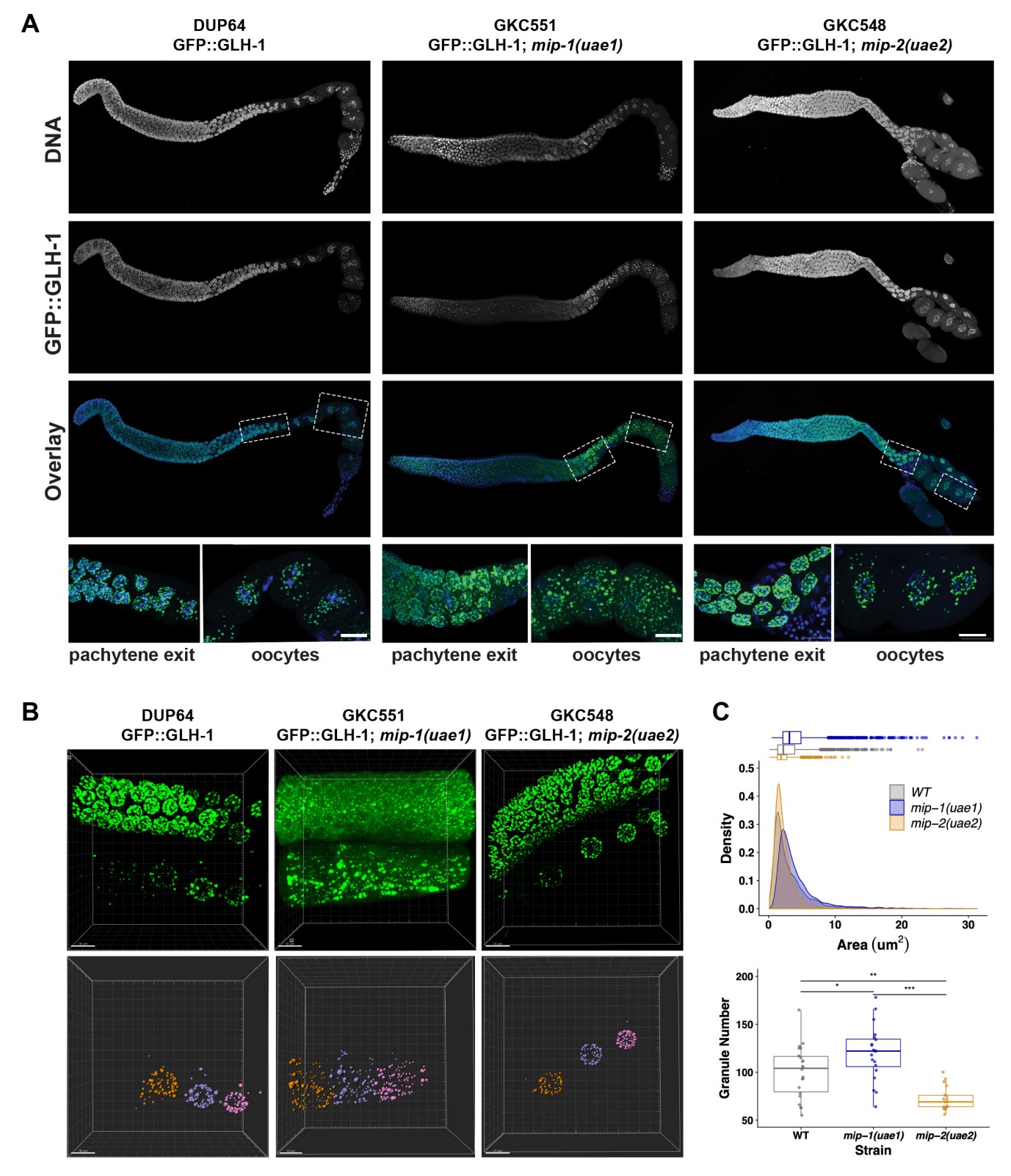

**Figure 10.** MIP-1 and MIP-2 are required for proper GLH-1 localization in vivo. (**A-B**) Localization of GLH-1::GFP in (**A**) fixed dissected gonads and (**B**) live animals of different genetic backgrounds: WT (strain DUP64), *mip-1(uae1)* null allele (strain GKC551), and *mip-2(uae2)* null allele (strain GKC548). (**A**) Granules show opposing trends in *mip-1* null (more cytoplasmic) and *mip-2* null (more perinuclear) backgrounds. Bottom row shows magnified views of boxed regions in overlays. (**B**) Representative examples used for Imaris quantification of granules in oocytes, showing inputs (top) and outputs (bottom).
*Figure 10 continued on next page*

*Figure 10 continued*

Granules are colored according to nuclear position relative to the uterus: −2 (orange), −3 (purple), −4 (fuchsia). A section of the meiotic germ line is visible above the oocytes. (C) Quantification of granule area (top) and number (bottom), combining data from the −2, −3, and −4 oocytes. Tukey's adjusted p-values from one-way ANOVA: *** < 0.001; ** 0.017; * 0.052.

The online version of this article includes the following video, source data, and figure supplement(s) for figure 10:

**Source data 1.** GLH-1 localization and granule size in the germline depends on MIP-1 and MIP-2.

**Figure supplement 1.** MIP-1 and MIP-2 are required for the proper localization of PGL-3 granules.

**Figure 10—video 1.** Localization of GLH-1 granules is affected in germ cells when individual *mips* are removed.

https://elifesciences.org/articles/60833#fig10video1

The MIP LOTUS domains show high predicted 3D structural similarity but little overall sequence conservation with LOTUS domains from other organisms, except in the hydrophobic core. Previous work in *Drosophila* has concluded that conservation of specific sequences at the Oskar LOTUS dimerization and Vasa binding interfaces is crucial for their interactions (*Jeske et al., 2015*; *Jeske et al., 2017*). Sequence alignments and 3D homology models, however, predict highly divergent interface contacts in the corresponding *C. elegans* complexes. Thus, it appears that a great

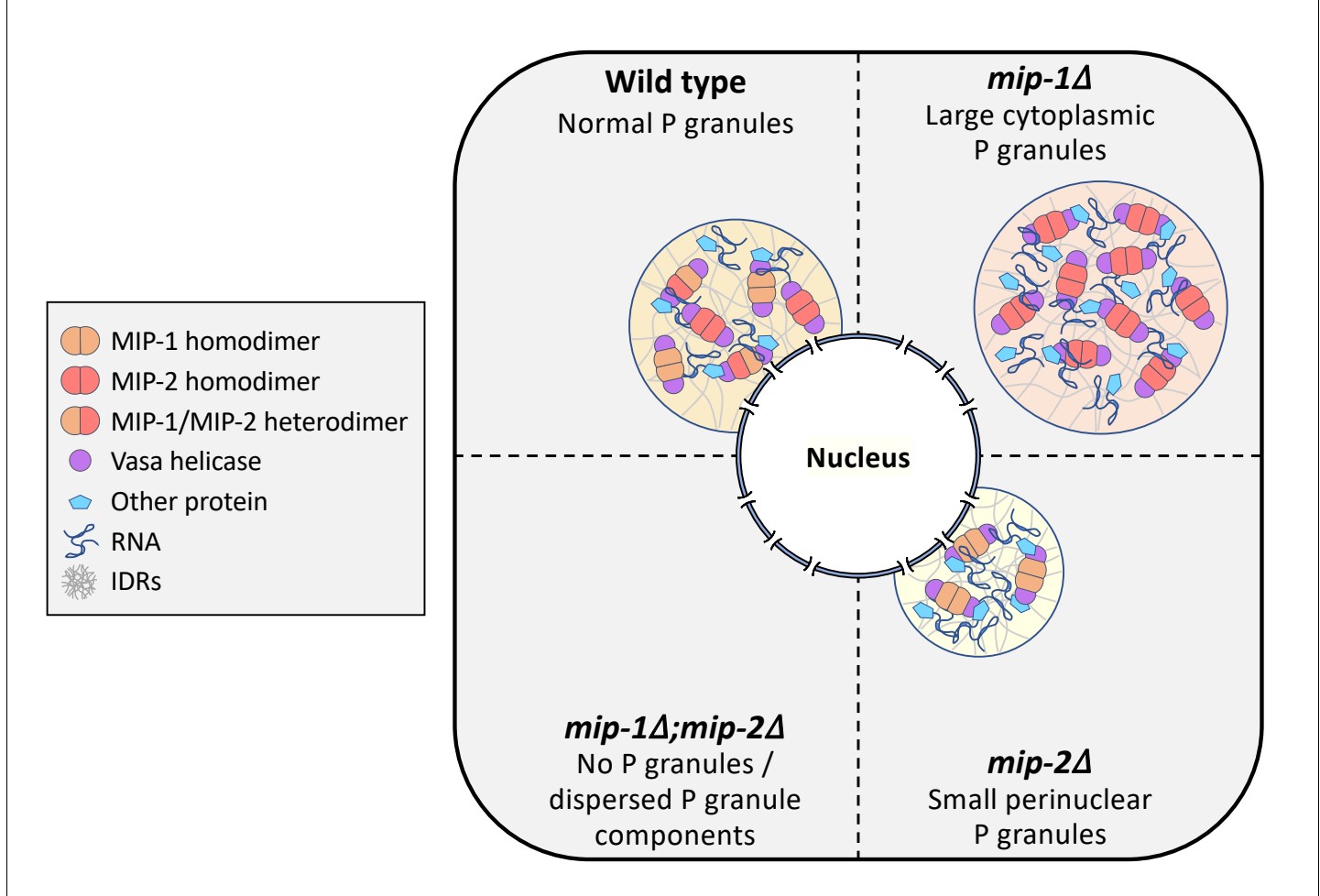

**Figure 11.** Conceptual illustration of MIP-1 and MIP-2 function in the *C. elegans* germ line. The MIPs form homo- and heterodimers and bind Vasa helicases through their LOTUS domains to nucleate and scaffold RNP complexes. These associations are likely enhanced by additional interactions between MIP IDRs and other proteins or RNAs. The MIPs balance each other to regulate P granule localization and size, and other P granule components fail to undergo phase separation when both are missing.

amount of plasticity in specific contacts can be tolerated while maintaining essential functional interactions.

Recent studies have highlighted the importance of multivalent proteins, which can engage several different partners, as hubs in the molecular networks underlying phase-separated compartments (*Sanders et al., 2020*; *Yang et al., 2020*). Taken together, our structure-function analyses support the idea that MIP-1 and MIP-2 act as hubs in an RNP interaction network that serve as a scaffold and recruitment platform to help organize molecular machinery within P granules. Since LOTUS domains and IDRs variously interact with other proteins or RNAs, further structure-function studies into how these contribute to phase separation and to the overall function of processes that occur within germ granules will be important to better understand their roles in vivo.

## Embryonic lethality and sterility

The *mip-1;mip-2* double null mutant gives rise to both embryonic lethality and progressive sterility that approaches 100% after only two generations at 25°C and elicits a mortal germ line (Mrt) phenotype within a few generations at 20°C. It is notable that many other P granule components also represent gene families containing 2–4 paralogs and display temperature-sensitive sterility. Such genes include *pgl-1* and *pgl-3* (*Kawasaki et al., 2004*; *Kawasaki et al., 1998*), *glh-1* and *glh-4* (*Spike et al., 2008a*), *meg-1* and *meg-2* (*Leacock and Reinke, 2008*), *meg-3* and *meg-4* (*Chen et al., 2016*; *Wang et al., 2014*), and *deps-1* (*Spike et al., 2008b*), which has a poorly annotated paralog in *C. elegans* (WormBase, http://www.wormbase.org, release WS276). Many of these proteins contain IDRs, and several form condensates in vitro – either alone or in the presence of RNA – or form granular structures upon expression in mammalian cells (*Chen et al., 2016*). *C. elegans* germ granules thus have a high level of built-in redundancy that is intrinsically linked with their biophysical properties and function.

The Mrt phenotype (*Ahmed and Hodgkin, 2000*) commonly occurs in association with defects in heritable RNAi and transgenerational epigenetic inheritance (TEI), which arise through misregulation of small RNA pathways (*Simon et al., 2014*; *Spracklin et al., 2017*) and downstream chromatin modifications (*Smelick and Ahmed, 2005*). Argonaute pathways use different classes of small RNAs to distinguish self from non-self and somatic versus germline mRNAs, and thereby either repress or license gene expression in the germ line (*Lee et al., 2012*; *Wedeles et al., 2013*). P granules are the primary site of piRNA and siRNA biogenesis in the *C. elegans* germ line, and several recent studies have described a role for them in balancing small RNA pathways (*Dodson and Kennedy, 2019*; *Lev et al., 2019*; *Ouyang et al., 2019*). Since different factors compete for some of the same machinery, stoichiometry of Argonautes and other small RNA pathway components is important not only for transgenerational memory, but also to tune the levels of developmentally important transcripts (*Dodson and Kennedy, 2019*; *Gerson-Gurwitz et al., 2016*; *Seth et al., 2018*).

Vasa helicases are essential for both mRNA regulation and small RNA processing in the germ line of many species (*Lasko, 2013*; *Voronina et al., 2011*). In both worms and flies, Vasa homologs associate with Argonautes and play a role in piRNA biogenesis and siRNA amplification (*Marnik et al., 2019*; *Xiol et al., 2014*). In *C. elegans*, GLH-1 also interacts with miRISC complexes within P granules to regulate translation and storage of target transcripts (*Dallaire et al., 2018*). Regulation of RNA helicase activity can either lock RNP complexes or remodel them by promoting RNA transfer between complexes, release, and turnover – all of which influence the composition, organization, and biophysical properties of condensates (*Marnik et al., 2019*). Thus, the dissolution of P granules upon loss of MIPs function is expected to have broad-ranging consequences on RNA metabolism.

## Germline development

Increasing sterility in *mip-1;mip-2* animals over successive generations is accompanied by a spectrum of progressively more severe phenotypes in the adult germ line, involving all stages of germline development: germline stem cell maintenance, meiotic progression, the sperm/oocyte switch, and gametogenesis. Specific phenotypes include germ lines that are nearly devoid of germ cells; pachytene arrest and failure to produce any mature gametes; production of sperm but no oocytes; or small and disorganized oocytes, indicating a failure of apoptosis and/or oocyte growth.

Spatiotemporal protein expression in the *C. elegans* germ line is primarily driven by post-transcriptional regulation (*Merritt et al., 2008*), and P granules positioned over nuclear pores facilitate

this by surveilling mRNAs as they exit germline nuclei (*Sheth et al., 2010*). Given that P granules are severely compromised in *mip-1;mip-2* mutants, their highly pleiotropic phenotypes likely reflect disruption of key post-transcriptional regulatory events. A complex network of RBPs, some of which localize to P granules, work together with the strongly conserved Notch and Ras-MAPK/ERK signaling cascades in a complex web of interactions to maintain totipotency and to coordinate developmental progression in the germ line (*Hubstenberger et al., 2012*; *Kisielnicka et al., 2018*). Mutations in different members of this network give rise to the full spectrum of phenotypes we observe upon loss of MIPs function, including defects in mitotic proliferation, meiotic progression, and sperm production, as well as feminization of the germ line and disorganized and aberrantly sized oocytes (*Kimble and Crittenden, 2007*; *Lee et al., 2007*; *Vaid et al., 2013*).

Occasionally, the *mip-1;mip-2* null strain gives rise to distal oocytes, resulting in apparent duplication of the oogenic program. This Gogo phenotype has also been observed in association with mutations in the *rpoa-2* gene, encoding RNA Polymerase I (*Eberhard et al., 2013*), and in *mina-1*, which encodes an RNA-binding protein present exclusively in the pachytene zone of the germ line (*Sendoel et al., 2019*). The *rpoa-2(op259)* mutation results in altered ribosomal RNA synthesis and confers resistance to germ cell apoptosis induced by ionizing radiation-induced DNA damage. MINA-1 is implicated in germline differentiation, physiological apoptosis and RNAi, and its absence also causes enlargement and disorganization of germline P granules. The second wave of oocyte differentiation in the distal gonad suggests that early meiotic cells begin to lose meiotic checkpoints sometime after a fully developed germ line has already been established in the adult hermaphrodite. While the proximal cause in *mip* mutants remains to be determined, the Gogo phenotype has been linked to alterations in the Ras/MAPK pathway, which directs developmental transitions in the germ line and is involved in regulating both physiological and DNA-damage-induced apoptosis (*Eberhard et al., 2013*; *Sendoel et al., 2019*).

The most extreme *mip-1;mip-2* phenotype is a greatly reduced germ line, containing only a few germ cells with abnormally large nuclei in close proximity to the somatic distal tip cell (DTC). The few remaining cells in atrophied *mip-1;mip-2* germ lines thus appear to have exhausted their capacity for self-renewal and mitotic proliferation. We do not yet know if they fail in asymmetric GSC divisions, have simply arrested the mitotic cell cycle and become quiescent, or if they degenerate after attempting meiosis. Since we never see distal tumors, these cells are not simply blocked at the point of meiotic entry. Enlarged nuclei can be caused by a variety of factors, including DNA damage; further analysis of this phenotype awaits future studies.

The broad range of *mip-1;mip-2* phenotypes we see suggests that cells in the germ line cannot properly maintain their state or manage state transitions upon prolonged absence of MIPs function. The precise timing of a misregulated developmental switch would determine whether cells become terminally arrested or make a decision to differentiate into sperm or oocytes. Further work will be needed to elucidate the mechanisms through which the MIPs influence specific developmental events in the germ line, whether these primarily involve post-transcriptional regulation and/or small RNA pathways, and whether the MIPs also interact directly with Notch and Ras/MAPK signaling cascades.

## P granule dynamics

MEG-3 nonspecifically recruits numerous mRNAs into P granules in vivo (*Lee et al., 2020*), and in vitro it forms molecular condensates with RNA, which lowers its saturation concentration (*Putnam et al., 2019*). Antagonistic regulation of MEG-3 by PP2A phosphatase and MBK-2 kinase, a master regulator of the oocyte-to-embryo transition, and competition for RNA with anteriorly localized MEX-5, create a permissive environment for P granule assembly only in the posterior of the zygote (*Brangwynne et al., 2009*; *Wang et al., 2014*). We do not yet know if the serine-rich MIPs are also directly targeted by MBK-2 and/or by other kinases.

The MIPs and MEG-3 have a synergistic relationship in P granule formation in the early embryo: when either MEG-3 or *mip-1*/*mip-2* are absent, other P granule components are mostly diffuse in the zygote. Therefore, the MIPs appear to work together with MEG-3 to decrease the critical concentration for P granule condensation. One model could be that MEG-3 initiates P granule formation after fertilization by recruiting RNAs from the bulk cytoplasm, and the MIPs promote their consolidation and growth by recruiting additional P granule components (proteins and potentially RNAs) to these nascent structures. It remains to be determined precisely how the interplay between the MIPs,

MEG-3, and other factors works to regulate the nucleation and growth of these complex coacervates.

P granules are dynamic structures that are variously cytoplasmic or perinuclear and which dissolve and reappear in a developmentally controlled manner. We found that PGL-3 was significantly more mobile than MEG-3, MIP-1, MIP-2, and GLH-1 in two-cell embryos, and that between the 2- and 12-cell stage its diffusivity slowed to match that of the other four proteins. PGL-3 therefore seems to become more firmly tethered within P granules in conjunction with their transition from free-floating to perinuclear structures, leading us to conclude that the biophysical properties of P granules in the embryo evolve over time.

MEG-3 has been described to form a ribbon-like structure surrounding and penetrating PGL granules in vivo (*Wang et al., 2014*) and to form an outer layer encasing a mobile PGL phase in reconstituted assemblies in vitro (*Putnam et al., 2019*). As visualized here by super-resolution imaging, MEG-3 appears to permeate the full volume of cytoplasmic P granules and is fully interspersed with MIP-2 at least through the eight-cell stage, suggesting that the higher mobility of PGL-3 in very early embryos occurs in the absence of spatially distinct, immiscible phases. Since multi-phasic compartmentalization is highly sensitive to valency and stoichiometry (*Banani et al., 2017*; *Peran and Mittag, 2020*; *Protter et al., 2018*; *Sanders et al., 2020*), it is possible that technical differences between studies – such as imaging techniques and in vitro vs. in vivo analyses – could lead to different conclusions about their architecture due to compositional heterogeneity under different conditions. Deconstructing the nature of these interactions will be an important focus for future study.

The MIPs play a central role in maintaining P granule integrity at all developmental stages. In contrast, embryos depleted of *meg-3* do eventually form P granules, since *meg-3* mutant larvae and adults show normal expression and localization of the MIPs and other P granule proteins in the germ line. Therefore, at least two distinct mechanisms exist to drive P granule assembly in vivo: nucleation and growth in the early embryo that depends on both MEG-3 and MIPs; and a MIP-dependent, MEG-3-independent process that nucleates and maintains P granules at the nuclear periphery later in germline development. This is consistent with the finding that phase separation of P granule components in the early embryo is not strictly required when other developmental determinants such as MEX-5 remain asymmetrically distributed in the zygote (*Gallo et al., 2010*).

In WT embryos, P granules begin attaching to the nuclear periphery in the P2 cell, when the P lineage is thought to be transcriptionally silent, and form caps over nuclear pores throughout the rest of germline development (*Pitt et al., 2000*). In the adult germ line, perinuclear granules assemble around RNA as it exits the nucleus, and transcription and mRNA export are required for their proper morphology (*Sheth et al., 2010*). Association with membranes can reduce the saturation concentration for condensation by localized recruitment of components and restricting diffusion (*Snead and Gladfelter, 2019*; *Söding et al., 2020*). If the primary role of MEG-3 is to recruit RNAs from the bulk cytoplasm, this would provide a plausible mechanism for MEG-3-independent assembly.

The perinuclear localization of MIP-1 and MIP-2 in the germ line, independent of other P granule components, raises the possibility that they nucleate the formation of P granules at the nuclear periphery. P granules likely attach directly to nuclear pore complexes (NPCs), as knockdown of some nuclear pore proteins causes them to detach (*Updike and Strome, 2009*; *Voronina and Seydoux, 2010*). Since MIP-1 typically appears closer to the base of perinuclear P granules than MIP-2 and is required for perinuclear localization of P granules, it is an obvious candidate for tethering granules to NPCs. The MIPs may work together with GLH-1 to maintain perinuclear association, since the FG repeats of GLH-1 are also necessary (but not sufficient) for its perinuclear localization in the embryo or when expressed ectopically (*Marnik et al., 2019*; *Updike et al., 2011*). The FG repeats are thought to help tether GLH-1 to nuclear pores through hydrophobic interactions with FG repeat proteins in the NPC matrix. In support of this idea, GLH-1 and PGL-1 readily diffuse upon treatment with 1,6-hexanediol (*Updike et al., 2011* and this study). In contrast, The MIPs are partially resistant to this treatment and therefore must employ additional interaction forces to engage other P granule and/or nuclear pore components. GLH-1 also depends on MIP-1 in addition to its FG repeats to help anchor it within P granules, since its mobility increases when MIP-1 is depleted. Future experiments will be needed to fully understand the mechanistic basis of perinuclear P granule association.

## Regulation of membraneless compartments

P granules and other biomolecular condensates are dynamic, non-equilibrium structures whose nucleation and growth are driven by a combination of high-affinity protein-protein and protein-RNA interactions and weak multivalent interactions between IDRs and with RNAs (*Banani et al., 2017*; *Hyman et al., 2014*; *Mittag and Parker, 2018*). Their biophysical properties are underpinned by a variety of molecular forces, including charge-charge, dipole-dipole, cation-pi, and pi-pi stacking interactions (*Brangwynne et al., 2015*). These interactions are influenced by solvent properties like pH and ionic strength, environmental factors such as temperature, and by the concentration, composition, and stoichiometry of their components. In vivo, the formation and organization of biomolecular condensates are regulated by cellular and developmental cues (*Söding et al., 2020*). High concentrations of free RNA and translational repressors tend to promote phase separation (*Langdon and Gladfelter, 2018*), whereas a wide variety of post-translational modifications (PTMs) – including Arg/Lys methylation, Lys acetylation, Arg citrullination, Ser/Thr and Tyr phosphorylation, Ser/Thr glycosylation, and SUMOylation – can either promote or antagonize LLPS by altering electrostatics and binding valency (*Brangwynne et al., 2015*; *Shin and Brangwynne, 2017*; *Snead and Gladfelter, 2019*). By changing the nature and strength of molecular interactions, PTMs thus provide a mechanism for rapid and reversible control of material properties that allow tunable dynamic responses to changing conditions.

RNA helicases also play active roles in remodeling biomolecular condensates by regulating RNP dynamics (*Linder and Jankowsky, 2011*). ATP-binding promotes condensation, and ATPase-coupled RNA duplex unwinding and release of ssRNA promote fluidity and RNA flux through phase-separated compartments (*Hondele et al., 2019*; *Hubstenberger et al., 2013*; *Marnik et al., 2019*; *Nott et al., 2015*). ATP-dependent Vasa helicases have long been associated with essential functions in the germ line, where they engage a variety of RNP complexes involved in translational regulation and small RNA biogenesis (*Dallaire et al., 2018*; *Lasko, 2013*). Vasa helicases such as GLH-1 therefore play a central role in coordinating numerous dynamic RNA regulatory processes within germ granules, where their activity would be expected to tune both biophysical properties and information flow. Since the MIPs directly bind and recruit the *C. elegans* Vasa homolog GLH-1 to P granules, many of the defects elicited in the absence of MIPs function could reflect a failure to properly organize different types of RNP reaction centers that require RNA helicase activity within them.

The broad spectrum of germline phenotypes we observe may be viewed in light of recent findings that biomolecular condensates enhance the robustness of switch-like cellular decisions and can also act as rheostats to tune outputs (*Klosin et al., 2020*). This is because phase separation selectively increases the efficiency of molecular reactions, both by concentrating specific molecular machinery and by excluding antagonistic regulators (*Snead and Gladfelter, 2019*; *Söding et al., 2020*). P granules provide a protected compartment for mRNA surveillance by small RNA pathways and translational regulators. Storage and regulation of transcripts encoding developmental determinants are promoted by a combination of mRNA stabilization and translational repression, as well as exclusion of assembled ribosomes (*Lee et al., 2020*; *Marnik et al., 2019*; *Parker et al., 2020*). Thus, while phase separation may not be absolutely required for the activity of individual P granule components, it generates a sequestered environment that helps buffer regulatory processes that contribute to the proper timing and execution of developmental transitions. The gradual appearance of wide-ranging developmental defects and consequent loss of reproductive potential in the absence of key germ granule components are therefore entirely consistent with the current view of how molecular condensates contribute to cellular robustness.

Membraneless organelles are complex dynamic assemblies with numerous constituents. To date, work on biomolecular condensates has primarily focused on the properties and roles of one or a small number of components in their assembly and dynamics. The identification of LOTUS-domain proteins that may organize and scaffold various RNP assemblies by recruiting a core Vasa helicase is a step forward toward understanding deeply conserved architectural principles of germ granules. Understanding their full range of molecular interactions, how their composition and dynamics are governed in vivo, and how these contribute to the regulation of essential cellular and developmental processes will be the next major challenge.

## Materials and methods

### Strains

*C. elegans* strains were maintained on NGM agar plates seeded with *E. coli* OP-50 as described previously (*Brenner, 1974*). Mutant strains were maintained at 20°C and fluorescently tagged strains at 22.5°C or 25°C (unless otherwise indicated). Fluorescently tagged MIP-1::GFP and MIP-2::mCherry strains were generated by Knudra Transgenics (Murray, UT; http://www.knudra.com/). All other strains were generated in our laboratory or obtained from other laboratories, either directly or through the *Caenorhabditis* Genetics Consortium (CGC; https://cgc.umn.edu/). *Supplementary file 1f* contains a comprehensive list of strains used in this study and their genotypes.

### Microscopy

Widefield imaging was performed on a Leica DM5500 B microscope with a Leica DFC365 FX camera using a 40x air objective or a 63X or 100X immersion oil objectives. Maximum projection images were produced using both the Leica LASX software and the ImageJ Bioformats Importer plug-in and represent between 30 and 60 z-slices unless otherwise noted. Super-resolution microscopy was performed using either a Leica SP8 with resonance scanner with HyD detectors or a Zeiss LSM880 with Airyscan using a 63X immersion objective (1.4 NA).

To quantify condensates in the adult germ line, P granule size and distance from the nucleus in WT and in *mip-1* or *mip-2* null mutant backgrounds were quantified in three dimensions using Imaris Image Analysis software ImarisCell. Granules (labeled 'vesicles' by Imaris) were detected using MIP-2::mCherry, MIP-1::GFP, and GLH-1::GFP signals. MIP analysis was performed using 10 fixed germ lines per genotype, which were stained with DAPI to detect DNA. GLH-1 analysis was performed using 6–8 live germ lines per genotype; these strains did not carry a DNA marker and hence distance from the nucleus was not quantified for those experiments. Oocyte cell boundaries were inferred automatically by Imaris and hand-corrected were needed. Measured distances from the nucleus and granule areas were exported for statistical analysis in R.

To quantify condensates in embryos, 10 dissected four-cell embryos per strain were imaged using the Zeiss LSM880 as indicated above. Scattering attenuation was adjusted using Zen Blue 2.3 Stack Correction (Background + Flicker + Decay). Condensates were quantified in the P2 cell using ImageJ 3D Object Counter.

### RNA interference

RNAi by feeding on solid medium was performed as previously described using clones from the Ahringer RNAi library (*Kamath et al., 2001*). To test for the effect of simultaneous knockdown of *mip-1* and *mip-2*, animals were treated with a 1:1 mixture of bacteria expressing dsRNA for *mip-1* (sjj_C38D4.4) and *mip-2* (sjj_F58G11.3) or empty vector control (L4440) using a continuous feeding protocol across three generations (see *Figure 2—figure supplement 1*). Briefly, 50 L1s (P0 generation) were plated in individual wells at 25°C for each treatment. After 24 hr, 8 P0 adults from each treatment were taken for live imaging. After another 24 hr, wells were scored for sterility, remaining P0 adults were removed from all wells, and 50 L1s (F1 generation) were then transferred into individual wells on new plates. Twenty-four hr later, the parental plates (from which P0 adults had been removed) were scored for embryonic lethality. This entire process was repeated for three generations (through scoring of F2 progeny). *Figure 2A and B* show combined results of experiments performed using strain OD95 and derivative strain GKC509.

### Scoring of sterility and embryonic lethality

Sterility of RNAi-treated animals was evaluated in each generation 48 hr after plating of L1 larvae, at which time treated animals were scored as sterile if neither larvae nor embryos were detected (unfertilized oocytes were not accounted for in the quantification). Embryonic lethality among the progeny of treated animals was evaluated 72 hr after L1s were plated (24 hr after removal of adults); embryonic lethality was scored in a binary fashion and was reported as present if over two thirds of the progeny in a well were unhatched embryos as determined by visual inspection (see *Figure 2—figure supplement 1*).

## Characterization of germline phenotypes

All germline defects detected in six experimental replicates (using two strains with and without the DTC marker) were characterized by visual inspection and compared with previously described RNAi phenotypes using the Phenotype Tool at WormBase (*Harris et al., 2020*). After extensive manual analysis, commonly observed phenotypes were grouped together to define broad phenotypic classes. All imaged gonads were then scored to quantify the distribution of phenotypes elicited in each generation.

## Germ line fixation and immunofluorescence

Germ line fixation and antibody staining were performed as previously described with some modifications (*Crittenden et al., 1994*). Briefly, 10–20 worms were placed on a coverslip in a 9 µM droplet of egg buffer containing 0.2 mM levamisole. To extrude the gonads, the animals' heads or tails were cut using two syringe needles. The coverslip was then carefully dropped onto a HistoBond microscope slide and placed on a cold block over dry ice to freeze. Once frozen, the coverslip was flicked off with a razor blade and immediately placed in 100% methanol at −20°C for 10 min, transferred to 100% acetone at −20°C for 5 min, and washed with an acetone dilution series (70%, 50%, 30% and 10%) at 4°C. Finally, the slide was transferred to PBS for 5 min.

For antibody staining, samples were blocked in PBST with 0.5% BSA and 0.5% non-fat milk by placing the solution directly on the slide and incubating in a humidity chamber for at least 1 hr at RT, after which this solution was replaced with primary antibody diluted in the blocking solution and incubated in a humidity chamber overnight at 4°C. The GLH-1 crude polyclonal antibody (Strome lab, UCSC, Santa Cruz, CA) was diluted at 1:500 and the monoclonal PGL-1 antibody (K76 – DSHB, University of Iowa) at 1:5. The slides were then washed three times in PBST for 5 min and incubated with TRITC-conjugated goat anti-rabbit (for α-GLH-1) or TRITC-conjugated goat anti-mouse (for K76) (Jackson ImmunoResearch laboratories Inc, West Grove, PA). Slides were incubated for 2 hr in a dark humidity chamber and then washed three times in PBST for 5 min. Finally, a coverslip with 5 µL of ProLong Glass with NucBlue was carefully placed on the sample and sealed with nail polish.

## CRISPR

The co-CRISPR strategy was used to generate all CRISPR strains produced in our laboratory (*Paix et al., 2017*). The online CHOPCHOP CRISPR design tool (http://chopchop.cbu.uib.no/) was used to design CRISPR guide (cr) RNAs. All CRISPR reagents (including the Cas9 protein, crRNAs, and ssDNA repair templates) were ordered from Integrated DNA Technologies, Inc (San Diego, California). We used the co-CRISPR marker *dpy-10*, which gives rise to F1 animals with a Rol or Dpy phenotype, to enrich for targeted CRISPR/Cas9-mediated genome editing events among the progeny. Injection and screening for genome-edited animals were performed as described (*Paix et al., 2017*). A list of repair templates and oligonucleotide sequences of guide RNAs for the strains generated can be found in *Supplementary file 1g*.

## Sequence and structural analysis

To characterize the MIP-1 and MIP-2 proteins and their interactions, we used sequence analysis (BLAST, PSI-BLAST, and multiple sequence alignment with PRALINE) (*Altschul et al., 1990*; *Altschul et al., 1997*; *Bawono and Heringa, 2014*; *Simossis and Heringa, 2005*) to identify conserved protein domains and used homology modeling to predict 3D domain structures, which were used to assemble and refine protein-protein complexes, and compute their binding affinities. These structural models were then used to guide interpretation of biological functions and to inform further experimental analyses.

For protein structure prediction, we used homology-based online servers including RaptorX (http://raptorx.uchicago.edu/StructurePrediction/predict/), I-TASSER (https://zhanglab.ccmb.med.umich.edu/I-TASSER/), and SWISS-MODEL (https://swissmodel.expasy.org/interactive). Homology modeling uses sequence alignment to detect related proteins and exploits experimentally solved template structures to build 3D models. Performance tests (CASP9,10) have shown that the RaptorX algorithm, the primary method used here, excels at predicting proteins with low sequence identity (<30%) (*Wang et al., 2016*). For each prediction, RaptorX reports measures of model quality,

including sequence identity, p-value, and global distance test (GDT), which scores predicted structures (values > 50 indicate good models) (*Källberg et al., 2012*).

To model protein-protein interactions, predicted *C. elegans* LOTUS and GLH-1 Vasa domains were assembled into LOTUS-LOTUS and LOTUS-Vasa-CTD complexes based on structural alignment using solved templates for the Oskar LOTUS domain (PDB ID: 5NT7). The complexes were then refined using the Monte Carlo Minimization (MCM) algorithm as implemented in Tinker molecular modeling package (*Rackers et al., 2018*). MCM performs iterative searches for conformations that lower the interaction energy by random sampling of the rotation angles of the protein backbone and side chains. Convergence of a complex is attained when no lower energy structures are found (typically within 10 days of run on a modern Linux cluster). Simulation parameters: temperature, 298K; convergence tolerance, 0.01 kcal/mol/Å; and maximum angle change for conformational sampling, 1.5°C.

Quantification of protein-protein interactions requires computation of their binding affinities, defined as the difference in interaction energies between the binary complex and its unbound components. We used standard molecular force fields (AMBER99) and Poisson-Boltzmann electrostatics to compute protein-protein interactions as implemented in Tinker and APBS packages, respectively; additionally, we considered entropic changes associated with macromolecular binding events (*Flamand et al., 2017*; *Gan and Gunsalus, 2013*; *Gan and Gunsalus, 2019*). To test this method, we computed, from the crystal structures (5NT7), the binding affinities of the Oskar LOTUS-LOTUS and LOTUS-Vasa-CTD complexes, which yielded −54.5 kcal/mol and −53.5 kcal/mol, respectively. Iterative titration calorimetry (ITC) experiments suggest that the LOTUS-Vasa-CTD association has a $K_D$~10 µM (*Jeske et al., 2017*), whereas gel experiments imply that LOTUS-LOTUS has a $K_D$ in the range of nM (*Jeske et al., 2015*). Thus, both experiment and modeling suggest the Oskar LOTUS-LOTUS dimer likely binds with a greater affinity than the Oskar LOTUS-Vasa-CTD complex, although a precise comparison requires more accurate measurements.

## Embryo affinity pull-down assays

Embryo affinity pull-down assays were performed using a modified protocol based on our previous study (*Chen et al., 2016*). Embryos (~2 million per replicate) were freshly harvested in biological triplicate by bleaching young gravid hermaphrodites and sonicated on ice (cycle: 0.5 s, amplitude: 45%, five strokes/session, five sessions, interval between sessions: 30 s; UP200S ultrasonic processor (Hielscher Ultrasonics GmbH)) in lysis buffer (total final volume: ~700 µl; 50 mM Tris-HCl pH 7.4, 100 mM KCl, 1 mM MgCl$_2$, 1 mM EGTA, 0.5 mM DTT, 10% glycerol, protease inhibitor cocktail (Roche), 0.1% Triton X-100). After sonication, Triton X-100 was added up to 1% and the lysates were incubated with head over tail rotation at 4°C for 30 min, followed by centrifugation at 20,000 × *g* at 4°C for 20 min. Cleared lysate was then aspirated without disturbing the upper lipid layer and split by half into either the anti-GFP agarose beads or the control blocked beads (22 µl, Chromotek). After head over tail rotation at 4°C for 90 min, the beads were washed once with ice-cold buffer I (50 mM Tris-HCl pH 7.4, 100 mM KCl, 1 mM MgCl$_2$, 1 mM EGTA, 10% glycerol, 0.1% Triton X-100), followed by washing in buffer II (50 mM Tris-HCl pH 7.4, 100 mM KCl, 1 mM MgCl$_2$, 1 mM EGTA, 10% glycerol) and then in buffer III (1 mM Tris-HCl pH 7.4, 150 mM KCl, 1 mM MgCl$_2$). Thereafter, proteins were eluted twice by shaking in 50 µl of 8 M guanidinium chloride at 90°C, followed by ethanol precipitation overnight at 4°C.

Precipitated proteins were re-solubilized in 6M urea/2M thiourea buffer (10 mM HEPES pH 8.0). Then, proteins were reduced by dithiothreitol and alkylated by iodoacetamide in the dark at room temperature, followed by in-solution digestion sequentially using lysyl endopeptidase (Lys-C, Wako) for 3 hr and trypsin (Promega) overnight at room temperature as previously described (*Paul et al., 2011*). Peptides were desalted and purified by solid phase extraction in C$_{18}$ StageTips (*Rappsilber et al., 2003*).

## Liquid chromatography tandem mass spectrometry

Peptides were resolved on an in-house packed analytical column (inner diameter: 75 µm; ReproSil-Pur C$_{18}$-AQ 3 µm resin, Dr. Maisch GmbH) by online nanoflow reversed phase chromatography through an 8–50% gradient of acetonitrile with 0.1% formic acid (120 min). The eluted peptides were sprayed directly by electrospray ionization into the Q Exactive Plus Orbitrap mass

spectrometer (Thermo Scientific). Mass spectra were acquired in data-dependent mode using a top10 sensitive method with one full scan (scan range: 300 to 1700 $m/z$ (mass-to-charge ratio), resolution: 70,000 at $m/z$ 200, target value: $3 \times 10^6$) followed by 10 fragmentation scans via higher energy collision dissociation (HCD; resolution: 35,000 at $m/z$ 200, target value: $5 \times 10^5$, maximum injection time: 120 ms, isolation window: 4.0 $m/z$, normalized collision energy: 26%). Precursor ions of unassigned or +one charge state were rejected for fragmentation scans. Precursor ions already isolated for fragmentation were dynamically excluded for 30 s.

## Mass spectrometry data analysis

Mass spectrometry raw data were processed by MaxQuant software (version 1.4.1.2) (*Cox and Mann, 2008*). With the built-in Andromeda search engine (*Cox et al., 2011*). Spectral data were searched against a concatenated target-decoy database consisting of the forward and reverse sequences of WormPep release WS245 (27,368 entries), *E. coli* K-12 MG1655 proteome (4285 entries) and a list of 247 common contaminants. In silico digestion of the proteome database was based on trypsin/P specificity (cleave C-terminally to arginine or lysine residues even if followed by proline) and a maximum of 2 missed cleavages. Carbamidomethylation of cysteine was chosen as fixed modification. Oxidation of methionine and acetylation of the protein N-terminus were considered as variable modifications. Minimum peptide length was seven amino acids. At least one unique peptide was required for each protein group. 'Second peptides' option was activated. 'Match between runs' function was sued to transfer identifications between measurement runs within a maximum retention time window of 1 min. False discovery rate (FDR) was set to 1% for both peptide and protein identifications. For protein quantification, both the unique and razor peptides were included (*Nesvizhskii and Aebersold, 2005*). Label-free quantification (LFQ) was performed using the MaxLFQ algorithm (*Cox et al., 2014*). Minimum LFQ ratio count was set to one.

## Mass spectrometry statistical analysis

Statistical data analysis was performed in the R statistical environment unless otherwise stated. Mass spectrometry data were analysed as in *Chen et al., 2016*. Briefly, proteins were filtered to retain those quantified in at least two out of the three GFP pull-down replicates. Next, LFQ intensities were log$_2$-transformed and the missing intensity values were imputed in Perseus software (version 1.2.0.17) (*Tyanova et al., 2016*) by random picking from a normal distribution that simulates low intensity values below the noise level (width = 0.3; shift = 1.8). The LFQ abundance ratio was then calculated for each protein between the GFP pull-downs and the controls. Significance of the enrichment was measured by an independent-sample Student's $t$ test assuming equal variances. Specific interactions were then determined in a volcano plot using a combined threshold (hyperbolic curve) based on the SAM (significance analysis of microarrays) algorithm ($s_0$ = 1.5, $t_0$ = 1.1 ~ 1.3) (*Tusher et al., 2001*). Proteins cross-reactive to the anti-GFP antibody were identified by pull-down experiments using the wild-type N2 strain. These cross-reactive proteins have been filtered out from the final interaction dataset.

## In vitro expression constructs

To express the full proteins and protein fragments with either the glutathione S-transferase (GST) or Histidine tag (His) in *E. coli,* total RNA was extracted from *C. elegans* embryos using a RNeasy Extraction Kit (Qiagen) and converted into cDNA using SuperScript IV Reverse Transcriptase (Invitrogen). Gene specific primers with restriction sites added to the 5' end were used to amplify the full CDS or gene fragments using Q5 DNA Polymerase (NEB). Primer sequences and restriction sites used for each construct can be found in *Supplementary file 1h*. The amplified PCR product was then cut using restriction enzymes (NEB) and ligated to linearized vector pGex-6p-1 (GST) or pET28-SUMO (His) also cut with the same restriction enzymes using T4 DNA ligase (NEB).

## Protein expression and purification

All recombinant proteins were expressed in either BL21-CodonPlus (DE3)-RIPL or ArcticExpress (DE3) competent cells (Agilent Technologies) grown in LB medium overnight at 13–16°C. For GST-recombinants, the cells were lysed using a sonicator (Fisher Scientific) in GST-lysis buffer (10 mM Tris-HCl, pH 8.0, 150 mM NaCl, 1 mM EDTA) supplemented with lysozyme (500 µg/mL), 1% sarkosyl

(sodium lauroyl sarcosinate), 1% Triton-X100 and protease inhibitor cocktail (Sigma). For His-tagged recombinants, the cells were lysed in His-lysis buffer (20 mM sodium phosphate, 0.5M NaCl, 20 mM Imidazole, 10% glycerol) supplemented with lysozyme, Triton-X100 and protease inhibitor cocktail. The protein was purified from cleared cell lysate using Ni-sepharose 6 Fast flow resin (GE Healthcare) in a poly-prep column (Bio-Rad). Following multiple washing steps with His-lysis buffer containing 60 mM imidazole, protein was eluted in His-lysis buffer containing 250 mM imidazole. Each fraction was analyzed in SDS-PAGE and Coomassie staining. Pure fractions were then concentrated using 50 k centrifugal filter units (Amicon).

## GST pull-downs

Approximately 5 µg of GST or GST-tagged protein fragments were incubated in Glutathione-Sepharose beads (GE Healthcare) in GST-lysis buffer at 4°C overnight. The bead-bound proteins were then incubated in GST-lysis buffer containing 5% BSA at 4°C for 2 hr. Meanwhile His-tagged full-length protein (~50 µg) was pre-cleared in Glutathione-Sepharose beads. The pre-cleared protein was then incubated with bead-bound proteins in Phosphate Buffered Saline containing 0.1% Tween 20 (PBST). After 2 hr incubation, the beads were washed three times with PBST containing 500 mM KCl and eluted with 2x-SDS loading buffer. The pull-downs were then analyzed by SDS-PAGE and subsequent Coomassie staining. For specific detection of His-tagged proteins, anti-His (1:1000) (Abcam) was used. Bound primary antibodies were detected using Goat anti-Mouse IRDye (1:10000) using an odyssey imaging system (Li-Cor).

## Yeast two-hybrid

Full-length or truncated cDNAs of *mip-1, mip-2,* and *glh-1* with DNA encoding 3×FLAG tag were cloned into the BamHI/SalI site of pGBD-C1 and the SalI/PstI site of pGAD-C1 (*James et al., 1996*). Yeast cells from strain PJ69-4a were transformed with 1 µg of plasmid DNA with carrier DNA and lithium acetate (*Gietz and Woods, 2006*). Transformed cells were then plated on the appropriate drop-out media to check for the presence of the plasmids. Tests for bait-prey interaction were done on SC-Leu-Trp-Ade plates or SC-Leu-Trp-His plates supplemented with 20 mM 3-amino-1,2,4-triazole.

## Mortal germ line assay

Double *mip* mutant animals were assessed for mortal germ line (Mrt) phenotype as described (*Ahmed and Hodgkin, 2000*), with minor modifications. For each plate or line, six L4 animals were transferred to a fresh plate every generation at 20°C. For these experiments, the generation number corresponds to the generation after thawing the line from our frozen stock. The number of fertile parent animals (animals with embryos in the uterus) was assessed after 48 hr by visual inspection under a dissecting microscope. The parents were removed and after 24 hr the number of live progeny as well as the number of unhatched embryos per plate were quantified. From these numbers, the average brood size and the percentage of embryonic lethality/progeny survival were calculated for every generation. Plates were scored as 'sterile' and the line stopped being followed when the brood size consisted of less than two progeny per fertile worm.

## Dispersal of GFP granules by alcohol treatment

1,6-Hexanediol treatment of dissected germ lines was performed following a modified protocol from a previously described report (*Updike et al., 2011*). Genome-edited worms expressing GFP-tagged versions of MIP-1, MIP-2, and GLH-1 were dissected in 8 µL droplets of egg buffer on poly-L-Lysine-coated coverslips. Images were acquired on an inverted Zeiss LSM880 microscope using the 63x objective every 10 s for a total time of 190 s. At the 20 s time point, 2 µL of either Egg buffer or 25% 1,6-hexanediol in Egg buffer (to reach a final concentration of the alcohol of 5%) were added to the droplet. Ten repeats were acquired for each of the three worm strains.

## Fluorescence recovery after photobleaching (FRAP) assay

All images were acquired on a Zeiss LSM880 confocal microscope using a 63x objective lens with NA of 1.4. Photobleaching of P granule proteins in live embryos was performed using a 488 nm Argon laser at 100% power to bleach an entire, single P granule. Post-bleaching, the granule was

continuously imaged with a 365 millisecond frame time for a recovery time of 60 s using a 488 nm laser power of 0.5% and a detector gain of 700V. All FRAP-related calculations were performed using the Zeiss FRAP module for ZEN Black.

## Contact for reagent and resource sharing

Further information and requests for resources and reagents should be directed to and will be fulfilled by the lead contact, Kristin Gunsalus (kcg1@nyu.edu).

## Acknowledgements

We thank Dr. K Kionte (NYU, New York) for help with the production of CRISPR lines; P Emhardt (NYUNY), F Mohammed, and S Gopinadhan (NYUAD) for technical assistance; Sean West for initial LOTUS domain identification; Rachid Rezgui (NYU Abu Dhabi Core Technology Platforms group) for help with Leica super-resolution microscopy; and Alan Twaddle and the NYU High Performance Computing team for computational support. We thank the members of the Gunsalus and Piano labs and Liz Marnik and Dustin Updike for helpful discussions and Tiffany Kilfeather for help with manuscript and figure preparation. This work was supported by a gift from NYU Abu Dhabi (to FP) and grants from the NYU Abu Dhabi Research Institute for the NYUAD CGSB (ADHPG CGSB to KCG), the Canadian Institutes of Health Research (CIHR) MOP 123352 (TFD), and the Charlotte and Leo Karassik Foundation (Ph.D. fellowship award to VKM). JXC received funding from the MDC-NYU PhD exchange program, supported by the BMBF (0315362) and the Berlin Institute for Medical Systems Biology (BIMSB). Some strains were provided by the CGC, which is funded by NIH Office of Research Infrastructure Programs (P40 OD010440).

## Additional information

### Funding

| Funder | Grant reference number | Author |
|---|---|---|
| New York University Abu Dhabi | ADPHG CGSB | Patricia Giselle Cipriani<br>Hala Fahs<br>Fabio Piano<br>Kristin C Gunsalus |
| New York University Abu Dhabi | | Patricia Giselle Cipriani<br>Olivia Bay<br>John Zinno<br>Michelle Gutwein<br>Hin Hark Gan<br>George Chung<br>Fabio Piano<br>Kristin C Gunsalus |
| Canadian Institutes of Health Research | MOP 123352 | Vinay K Mayya<br>Thomas F Duchaine |
| McGill University | Ph.D. Fellowship Award | Vinay K Mayya |
| Bundesministerium für Bildung und Forschung | 0315362 | Jia-Xuan Chen |
| NIH | P40 OD010440 | George Chung |

The funders had no role in study design, data collection and interpretation, or the decision to submit the work for publication.

### Author contributions

Patricia Giselle Cipriani, Conceptualization, Data curation, Supervision, Validation, Investigation, Methodology, Writing - original draft, Project administration, Writing - review and editing; Olivia Bay, Data curation, Validation, Investigation, Visualization, Writing - review and editing, Imaging, phenotypic analysis, and hexanediol experiments; John Zinno, Validation, Investigation, Visualization, Methodology, Writing - review and editing, Carried out imaging and performed and analyzed FRAP

experiments; Michelle Gutwein, Data curation, Software, Formal analysis, Investigation, Visualization, Methodology, Writing - review and editing, Designed CRISPR constructs, cloned expression constructs, analyzed data of multiple experiments, and produced graphs for data visualization; Hin Hark Gan, Software, Formal analysis, Investigation, Visualization, Methodology, Writing - review and editing, Sequence analysis and 3D modelling; Vinay K Mayya, Validation, Investigation, Visualization, Methodology, Writing - review and editing, Cloned expression constructs and performed GST-pull down experiments; George Chung, Validation, Investigation, Visualization, Methodology, Writing - review and editing, Designed Y2H constructs and performed Y2H experiments; Jia-Xuan Chen, Software, Formal analysis, Investigation, Visualization, Methodology, Writing - review and editing, Performed and analysed the initial co-IP experiments with MEG-3 and proposed candidate P granule proteins for follow-up studies; Hala Fahs, Investigation, Visualization, Methodology, Writing - review and editing, Designed CRISPR constructs, performed CRISPR genome editing and microscopy; Yu Guan, Data curation, Validation, Investigation, Visualization, Methodology, Writing - review and editing, Performed imaging and phenotypic analysis and quantified the Mrt phenotype; Thomas F Duchaine, Supervision, Funding acquisition, Writing - review and editing, Supervised pull-down experiments; Matthias Selbach, Supervision, Funding acquisition, Writing - review and editing, Supervised the proteomics experiments; Fabio Piano, Conceptualization, Supervision, Funding acquisition, Writing - review and editing, Contributed to conceptualization and supervision; Kristin C Gunsalus, Conceptualization, Supervision, Funding acquisition, Methodology, Writing - original draft, Project administration, Writing - review and editing

## Author ORCIDs

Patricia Giselle Cipriani (iD) https://orcid.org/0000-0001-9772-1012
Jia-Xuan Chen (iD) https://orcid.org/0000-0002-3671-553X
Kristin C Gunsalus (iD) https://orcid.org/0000-0001-9769-4624

## Decision letter and Author response

Decision letter https://doi.org/10.7554/eLife.60833.sa1
Author response https://doi.org/10.7554/eLife.60833.sa2

## Additional files

### Supplementary files

• Supplementary file 1. An Excel file containing multiple tables in different tabs. (a) Table S1. Proteins enriched in GFP::MEG-3 pulldowns. Proteins are sorted according to their p-value from the $t$ SAM statistic as previously described (**Chen et al., 2016**). (b) Table S2. Similarity measures between predicted MIP LOTUS domain structures and LOTUS domains from other metazoans. Protein domains: M1, MIP-1; M2, MIP-2; L1, LOTUS1; L2, LOTUS2. SeqID, sequence identity; GDT, Global Distance Test parameters for best templates; PDB ID, protein databank identifier of the best template; RMSD, backbone root mean square distances in Å, with number of aligned residues in parentheses. Published structures used for comparison: *D. melanogaster* Oskar (PBD ID 5nt7), *H. sapiens* TDRD5 (PBD ID 3s93). (c) Table S3. Pairwise MIP LOTUS structural similarity analysis. Values shown are backbone rmsd values in Å and number of aligned residues (in parentheses). M1, MIP-1, M2, MIP-2; L1, LOTUS1; L2, LOTUS2. (d) Table S4. Description of MIP depletion phenotypes. (e) Table S5. Pairwise predicted binding affinities between MIP LOTUS domains and between MIP LOTUS domains and GLH-1. Affinities are in kcal/mol. M1, MIP-1; M2, MIP-2; L1, LOTUS1; L2, LOTUS2. Predictions for GLH-1 binding considered only the helicase CTD domain. Predictions for combinations with no values given were highly unfavorable (>0 kcal/mol). Predicted binding affinities for the native *Drosophila* complexes: Oskar LOTUS homodimer = −54.5 kcal/mol; Oskar LOTUS—Vasa helicase complex = −53.5 kcal/mol. (f) Table S6. Strains produced and used in this study. (g) Table S7. Guide RNA sequences, repair templates, and screening primers for CRISPR strains produced in this study. (h) Table S8. Plasmid DNA constructs for in vitro pulldown experiments.

• Transparent reporting form

## Data availability

All mass spectrometry raw data have been deposited to the PRIDE repository with the dataset identifier PXD012852. All other data generated or analyzed during this study are included in the manuscript and supporting files. Source data files have been provided for Figures 2A–C; Figure 2—figure supplement 2; Figure 6A,B; Figure 8E; Figure 9B; Figure 9—figure supplement 1; Figure 10C.

The following dataset was generated:

| Author(s) | Year | Dataset title | Dataset URL | Database and Identifier |
|---|---|---|---|---|
| Cipriani PG, Bay O, Zinno J, Gutwein M, Gan HH, Mayya V, Chung G, Chen J-X, Fahs H, Guan Y, Duchaine T, Selbach M, Piano F, Gunsalus KC | 2021 | Novel LOTUS-domain proteins are organizational hubs that recruit *C. elegans* Vasa to germ granules | https://www.ebi.ac.uk/pride/archive/projects/PXD012852 | PRIDE, PXD012852 |

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
