## [Decision Letter]

**Acceptance summary:**

Further understanding of germ granule formation will have a significant impact on the field of germ cell biology. Using a series of biochemical, genetic, and cell-imaging experiments, this study significantly advances our understanding of germ granules by describing how two previously uncharacterized *C. elegans* proteins, MIP-1 and MIP-2, participate in the formation and regulation of these condensates. These findings provide important new insights into the biology of germ granules and the regulation of liquid-liquid phase separation during germ cell development.

**Decision letter after peer review:**

Thank you for submitting your article "Novel LOTUS-domain proteins are organizational hubs that recruit *C. elegans* Vasa to germ granules" for consideration by *eLife*. Your article has been reviewed by 3 peer reviewers, including Michael Buszczak as the Reviewing Editor and Reviewer #1, and the evaluation has been overseen by Piali Sengupta as the Senior Editor.

The reviewers have discussed the reviews with one another and the Reviewing Editor has drafted this decision to help you prepare a revised submission.

Summary:

Cipriani et al. present the identification of two LOTUS domain proteins that immunoprecipitate and co-localize with the P-granule component MEG-3. These phase-separating molecules affect the localization of a subset of other P granule components and physically interact with the GLH-1 helicase by Y2H and pull-down. The combination of genetic, biochemical, and modeling studies of novel P granule components make this a potentially attractive manuscript. However, there are substantial concerns about the figures and interpretation and clarity of writing that would need to be addressed for publication.

Essential revisions:

1) Most critically, the paper is based on the premise that the IPs identify new P granule components, however the data to support this claim are buried in in Suppl Figure S3A and S4. These need to be moved to main figures in the manuscript. It should be noted that the image in Figure S3A shows partial colocalization, at best, with PGL-1 – instead, they appear as mostly adjacent, rather than overlapping, proteins. Furthermore, how do the authors explain the subset of particles that do not include PGL-1? Colocalization in the adult germ line also needs to be shown. Lastly, several images throughout the paper (i.e. panels in Figure 5 and 8) appear dim and difficult to evaluate. Brighter and higher magnification images should be to shown and/or the authors should consider using pseudo coloring that offers better contrast. Appropriate controls should also be included in the figures (see specific comments below for details)..

2) Unless more substantively validated, the modeling experiments should be moved to the supplement (or removed) and text describing their predictive value should be softened. For example, the modeling of the Lotus domain in Figure S1B-C seems unnecessary and the reviewers are concerned about the rigor of comparing the predicted *C. elegans* Lotus structures to *Drosophila* Oskar and other solved structures. Especially when this data is used to make the conclusion that the differences between the *C. elegans* Lotus domain "structures" suggests they perform separable functions.

3) While the authors describe the phenotypic consequence of loss of MIP-1/2, they do not connect these phenotypes to known defects in other P granule components. The authors' conclusions about the dependence of known P granule components on mip-1 and mip-2 appear overstated as the images clearly show granular components in some of these mutants. Could it be that MIP-1/2 are required for the posterior localization or retention of the granules, not the formation per se? The section on the interdependence of known P granule components and the MIPs is poorly organized and therefore difficult to read. They would be better addressing the localization of all other components in the mip-1/mip-2 backgrounds all together and then MIP-1/2 proteins in all of those backgrounds together.

---

## [Author Response]

1) Most critically, the paper is based on the premise that the IPs identify new P granule components, however the data to support this claim are buried in in Suppl Figure S3A and S4. These need to be moved to main figures in the manuscript. It should be noted that the image in Figure S3A shows partial colocalization, at best, with PGL-1 – instead, they appear as mostly adjacent, rather than overlapping, proteins. Furthermore, how do the authors explain the subset of particles that do not include PGL-1? Colocalization in the adult germ line also needs to be shown. Lastly, several images throughout the paper (i.e. panels in Figure 5 and 8) appear dim and difficult to evaluate. Brighter and higher magnification images should be to shown and/or the authors should consider using pseudo coloring that offers better contrast. Appropriate controls should also be included in the figures (see specific comments below for details)..

We have combined the data from our original Figures S3 and S4 and have included additional panels to show the colocalization of the MIPs with the P granule components PGL-1 and MEG-3 in a new main figure (Figure 4). The image of an embryo previously shown in the original Figure S3 appeared to the reviewer to show adjacent localization due to the level of resolution of the image. The new main Figure 4 includes newer, higher-magnification panels showing more clearly the colocalization between the MIPs and PGL and between MIPs and MEG-3, in both the embryo as well as the adult germ line.

As previously shown for MEG-3 in Wang et al. 2014, some granules in the embryo (see Figure 4A) do not include PGL-1. These granules are generally localized toward the anterior, where it has been previously hypothesized that PGL granules are disassembled faster than other components like MEG-3 (Wang et al., 2014). We now include these observations in the text.

2) Unless more substantively validated, the modeling experiments should be moved to the supplement (or removed) and text describing their predictive value should be softened. For example, the modeling of the Lotus domain in Figure S1B-C seems unnecessary and the reviewers are concerned about the rigor of comparing the predicted *C. elegans* Lotus structures to *Drosophila* Oskar and other solved structures. Especially when this data is used to make the conclusion that the differences between the *C. elegans* Lotus domain "structures" suggests they perform separable functions.

Although we have had very good success in the past modeling protein complexes with high predictive ability and good agreement with experimental data (Gan and Gunsalus, NAR 2013; Gan and Gunsalus, NAR 2015; Flamand et al., NAR 2017; Chahal et al., NAR 2019; Gan et al., JMB 2021), we agree that homology models provide only a preliminary analysis of potential 3D structures and complexes and have revised and softened the related text in the Results accordingly. To avoid placing too much emphasis on specific RMSDs, we have also removed the rest of the paragraph describing the structural comparisons and now say simply, “Superpositions of LOTUS domain models from worm, fly, and human display clear structural similarity (Supplementary File 1b,1c).”

Regarding Figure S1 (Figure 1—figure supplement 1), the basic message that we want to convey is that the combination of multiple sequence alignments, secondary structural predictions, and tertiary structural models provides clear evidence that the overall structural similarity between the MIP and Oskar LOTUS domains is remarkably high, despite the fact that the overall sequence conservation is very low. The models in Figure S1B and C (Figure 1—figure supplement 1B,C) provide visual aids to understand the basis of this similarity, and we therefore believe there is value in retaining them. Specifically, Figure S1B (Figure 1—figure supplement 1B) illustrates that the pattern of sequence conservation shown in Figure S1A (Figure 1—figure supplement 1A) maps to the hydrophobic core of the domain and thus reveals the likely reason why these specific residues are so strongly conserved. The model comparisons in Figure S1C (Figure 1—figure supplement 1C) illustrate that we are able to identify the α 5 helix in all four different LOTUS domains, which is a distinctive feature of the extended LOTUS (eLOTUS) sub-class of this domain family. While a definitive determination will require solving the structures of these domains directly, the models provide additional support, beyond secondary structural predictions, for classifying these as eLOTUS domains instead of mLOTUS domains and will help direct future studies on their function. We believe that sharing all of the evidence we gathered on the potential functions of these domains will be useful for the community.

3) While the authors describe the phenotypic consequence of loss of MIP-1/2, they do not connect these phenotypes to known defects in other P granule components. The authors' conclusions about the dependence of known P granule components on mip-1 and mip-2 appear overstated as the images clearly show granular components in some of these mutants. Could it be that MIP-1/2 are required for the posterior localization or retention of the granules, not the formation per se?

We now point out in the manuscript that the granule phenotypes observed upon depletion of both MIPs in early embryos, as revealed by localization defects in other P granule components, are very similar to those previously described as a consequence of the depletion of MEG-3 and -4 (Wang et al. 2014; Ouyang et al. 2020).

Regarding the presence of residual granules, please note that since the double *mip* deletion lines rapidly become sterile, we were not able to maintain fluorescently tagged versions of other P granule proteins in this genetic background. Therefore, these experiments were performed using double *mip-1;mip-2* RNAi, which results in some variability in the amount of residual protein detected (and likely a weaker effect on P granule formation). To better illustrate our observations, we have replaced many of the micrographs obtained by epifluorescence with new images obtained with the Zeiss LSM880 confocal microscope (Figure 5A). The improved sensitivity and resolution of this instrument allowed us to better detect and visualize the localization of residual condensates in the embryo. These experiments confirmed that P granule assembly is indeed compromised, particularly at the earliest stages.

We have also added new time-lapse videos taken with this instrument showing examples of how these residual granules behave through the first two cell divisions upon *mip-1(RNAi);mip2(RNAi)* (Videos 1-3). In the course of repeating these experiments, we noticed that the intensity and apparent size of the residual granules can vary considerably depending on the brightness of the fluorophores in different strains, even for the same tagged protein. This is illustrated by Video 2, which was acquired using a brighter GLH-1::GFP strain in which residual granules are more prominent than those we observed previously.

Prompted by the reviewer’s comment regarding posterior retention of granules, we also examined this question more closely. This reanalysis showed that PGL-3, but not GLH-1 or MEG-3, additionally fails to concentrate in the posterior region of P lineage cells prior to mitosis and thus becomes mis-segregated to the sister cell. Therefore, the posterior localization of some, but not all, P granule components does not occur normally in the absence of the MIPs.

Finally, we have not found a previous report in the literature of another pair of paralogs that show opposing effects on the size and perinuclear attachment of granules in the germ line, as we describe for the MIPs in this manuscript. As requested, to better demonstrate this effect we have quantified the size and spatial distribution of granules in different genetic backgrounds. These data are now presented in a new main figure (Figure 9, MIP granules) and in additional panels added to a previous figure (Figure 10, GLH-1 granules).

We have revised the text in the manuscript to include all of the new observations resulting from these experiments.

The section on the interdependence of known P granule components and the MIPs is poorly organized and therefore difficult to read. They would be better addressing the localization of all other components in the mip-1/mip-2 backgrounds all together and then MIP-1/2 proteins in all of those backgrounds together.

We have rearranged the description of this section to make it clearer and revised the associated figure accordingly (Figure 5).